# KIF16B drives MT1-MMP recycling in macrophages and promotes co-invasion of cancer cells

Sven Hey , Christiane Wiesner , Bryan Barcelona , Stefan Linder

**The matrix metalloproteinase MT1-MMP is a central effector of cellular proteolysis. Accordingly, regulation of the surface-localized pool of MT1-MMP is crucial for cell migration and invasion. Here, we identify the superprocessive kinesin KIF16B as a major driver of fast recycling of MT1-MMP to the surface of primary human macrophages. KIF16B associates with MT1-MMP on Rab14-positive vesicles, and its depletion results in strongly reduced MT1-MMP surface levels, as shown by microscopical, biochemical, and cell-sorting approaches. As a consequence, KIF16B-depleted macrophages exhibit strongly reduced matrix degradation and invasion. We further identify the cargo-binding C-terminus of KIF16B as a critical element of MT1-MMP transport, as its overexpression uncouples MT1-MMP vesicles from the endogenous motor, thus leading to a reduction of surface-associated MT1-MMP and to reduced matrix degradation and invasion. Importantly, depletion of KIF16B in primary macrophages also reduces the co-invasion of cancer cells from tumor spheroids, pointing to the KIF16B-driven recycling pathway in macrophages as an important regulatory element of the tumor microenvironment.**

## Introduction

Macrophages are a central part of the innate immune system and represent the first line of defense against invading pathogens (Lavin et al, 2015; Levin et al, 2016). By recognizing non-self cells, and by clearing cell debris, macrophages are also critically involved in tissue homeostasis (Lavin et al, 2015). In contrast, a high load of tumor-associated macrophages is correlated with poor patient prognosis (Condeelis & Pollard, 2006). This adverse effect is likely because of the establishment of paracrine loops between macrophages and cancer cells that stimulate tumor invasiveness (Wyckoff et al, 2004), but also on the generation of local defects in the ECM by macrophages (Wiesner et al, 2014), which support the escape of cancer cells from the primary tumor during metastasis (Dovas et al, 2013).

The ability of macrophages to degrade ECM material and to cross tissue boundaries is based on the localized release or surface exposure of respective enzymes, notably of matrix metalloproteinases (MMPs), a family of $Zn^{2+}$-dependent endopeptidases comprising both secreted- and membrane-associated isoforms (Itoh, 2015; Quintero-Fabian et al, 2019). One of the best-studied members is MT1-MMP/MMP14, which is anchored to the cell surface by a transmembrane region (Gifford & Itoh, 2019). MT1-MMP plays important roles in a variety of physiological and pathological scenarios, including leukocyte extravasation, inflammation, diabetes, and metastasis (van Hinsbergh et al, 2006; Barbolina & Stack, 2008; Savinov & Strongin, 2009; Castro-Castro et al, 2016; Quintero-Fabian et al, 2019). Among its substrates are ECM components such as collagens I, II, and III, fibronectin, laminins 1 and 5, and fibrin (Itoh & Seiki, 2004), cell-matrix receptors such as pro-αV- and α5 integrins (Barbolina & Stack, 2008), CD44 (Kajita et al, 2001) or syndecan-1 (Endo et al, 2003), and proforms of other MMPs such as MMP-2, -8, and-13 (Itoh & Seiki, 2004; Barbolina & Stack, 2008), placing MT1-MMP center stage in the protease web of the cell.

Accordingly, MT1-MMP surface exposure and activity are tightly regulated (Hey et al, 2022). Removal of the MT1-MMP prodomain by convertases such as furin can occur already at cargo vesicles, preceding delivery to the plasma membrane (Sato et al, 1996). Activity of MT1-MMP is regulated, among others, by tissue inhibitors of metalloproteinases or testican (Nakada et al, 2003), or by reversin-inducing cysteine-rich protein with Kazal motifs (RECK) (Itoh & Seiki, 2004), and also by oligomerization of the protease itself (Lehti et al, 2002). We could show earlier that intracellular trafficking is central for the regulation of the surface-associated pool of MT1-MMP in primary human macrophages, by demonstrating that MT1-MMP is transported in Rab8a-positive vesicles that are driven by kinesin-1 and -2 motor proteins (Wiesner et al, 2010, 2013). Furthermore, surface-associated MT1-MMP can be endocytosed in Rab5a-positive endosomes and subsequently re-exposed on the cell surface through Rab14-dependent fast recycling or Rab22a-regulated slow recycling pathways, with Rab14-controlled fast recycling being the predominant pathway in 3D settings (Wiesner et al, 2013). However, the motor protein that drives MT1-MMP recycling, as well as its relative importance to MT1-MMP surface homeostasis, remained elusive. We now identify KIF16B, a

Institut für Medizinische Mikrobiologie, Virologie und Hygiene, Universitätsklinikum Eppendorf, Hamburg, Germany

Correspondence: s.linder@uke.de

superprocessive kinesin (Hoepfner et al, 2005), as a major motor protein that drives Rab14-dependent fast recycling of MT1-MMP back to the macrophage surface.

KIF16B is a member of the kinesin-3 family (Hoepfner et al, 2005) and is widely expressed in tissues, including brain, heart, liver, kidney, placenta, skeletal muscle, and also in white blood cells (Li et al, 2020). Typical for kinesins, it contains an N-terminal motor domain, several coiled coil (CC) regions for dimerization, and a C-terminal cargo-binding domain (Hoepfner et al, 2005). Moreover, KIF16B shows a combination of unique features. First, KIF16B features a "stalk inhibition" mechanism, in which binding of its second and third CCs to the motor domain inhibits association with microtubules and thus controls transport activity (Farkhondeh et al, 2015). Second, dimerized KIF16B is a superprocessive motor with an average run length of ~10 $\mu$m, making it suitable for long-range intracellular transport (Soppina et al, 2014). Third, the C-terminal region of KIF16B contains a PhoX (PX) domain, which enables direct binding to the membrane lipid PI(3)P (Hoepfner et al, 2005). KIF16B is thus one out of only three kinesins known to bind lipids, in addition to KIF13B binding PI(3,4,5)$P_3$ (Horiguchi et al, 2006) and KIF1A binding PI(4,5)$P_2$ (Klopfenstein et al, 2002). Fourth, KIF16B interacts directly with the GTP-bound, i.e., active, form of Rab14, which has been shown to be critical for murine early embryonic development (Ueno et al, 2011).

In line with its superprocessive nature, its ability to bind PI(3)P, a lipid especially enriched at early endosomes (Stenmark & Gillooly, 2001), and its direct binding of Rab14, a RabGTPase involved in fast recycling (Sonnichsen et al, 2000), KIF16B is thus ideally poised to regulate the rapid traffic of endocytosed cargo back to the cell surface. However, so far, only the cell surface receptors FGFR (Ueno et al, 2011) and TfR (transferrin receptor) (Perez Bay et al, 2013), as well as MHC I (major histocompatibility complex) (Weimershaus et al, 2018), have been identified as cargo of KIF16B, leaving its role in other transport processes unclear (Li et al, 2020). We now identify MT1-MMP as a novel cargo for KIF16B and show that KIF16B is the motor responsible for endosomal recycling of MT1-MMP to the surface of human macrophages. Moreover, we demonstrate that the KIF16B-driven pathway in macrophages is critical for their matrix-degrading and invasive abilities, and notably also for co-invasion of cancer cells.

## Results and Discussion

### KIF16B associates with MT1-MMP in endosomal recycling pathways

To test a potential involvement of the kinesin KIF16B in the intracellular transport of MT1-MMP vesicles, both overexpressed and endogenous forms of the motor protein were visualized in primary human macrophages. Strikingly, live cell imaging of macrophages coexpressing KIF16B-YFP and MT1-MMP-mCherry revealed a high degree of colocalization at both stationary and saltatory vesicles (Fig 1A–C, D1–D6; Video 1). This colocalization was confirmed also for both endogenous proteins (Fig 1E–G). To assess the relative importance of KIF16B and also to map its localization in the trafficking pathways of MT1-MMP, we performed quantitative colocalization

studies. Of note, motor proteins and their vesicular cargo can be associated with different subdomains of vesicles, which is particularly evident by visualizing giant endosomes, which can occasionally be formed as a result of KIF16B overexpression (Fig S1A). We thus chose an object-based colocalization approach (Moser et al, 2017), which involves thresholding and watershedding steps during image processing, and results in object (i.e., vesicle) masks (see also Fig 3A).

This analysis revealed substantial colocalization between MT1-MMP-mCherry with both overexpressed KIF16B-YFP (55.7% ± 1.7%) and endogenous KIF16B (20.7% ± 1.4%). By contrast, the previously identified regulators of MT1-MMP transport (Wiesner et al, 2010), kinesin-1 (KHC: 4.4% ± 0.6%), and kinesin-2 (KIF3A: 2.0% ± 0.4%), showed significantly less colocalization (Fig 1H). Double overlaps of KIF16B-YFP with endogenous RabGTPases showed that this motor is present mostly at Rab14-positive vesicles (48.3% ± 6.0%) (Fig 1I), whereas triple overlaps showed that MT1-MMP-mCherry colocalizes with KIF16B mostly at compartments positive for Rab5a (23.8% ± 2.7%), Rab14 (26.8% ± 1.6%), and Rab22a (19.3% ± 2.6%), which, respectively, mark endocytic, and fast and slow recycling pathways (Zhen & Stenmark, 2015) (Fig 1J).

Collectively, these results show that KIF16B localizes at a significant subset of MT1-MMP-mCherry–positive vesicles, even to a higher degree than the previously identified regulators of MT1-MMP trafficking, kinesin-1 and -2 (Wiesner et al, 2010). Moreover, KIF16B colocalizes with MT1-MMP mostly in endocytic and recycling pathways, whereas kinesin-1 and -2 colocalize with MT1-MMP mainly in exocytic pathways. These data are also in line with earlier findings of KIF16B transporting early endosomes in a Rab5-dependent pathway in HeLa cells (Hoepfner et al, 2005), or of FGFR2 (fibroblast growth factor receptor 2) vesicles in a Rab14-dependent manner in embryonic cells (Ueno et al, 2011).

The finding that this colocalization rate is even higher for a KIF16B-YFP construct indicates that overexpression of the motor enlarges the MT1-MMP pool present in recycling pathways, which is also in agreement with earlier findings of KIF16B transporting early endosomes to microtubule plus ends in the cell periphery of HeLa cells (Hoepfner et al, 2005). Furthermore, our earlier measurements of a speed of ~0.35 $\mu$m/s for MT1-MMP vesicles in the cell periphery of macrophages (Wiesner et al, 2010) appear to be in accordance with the maximal gliding velocity of ~0.25 $\mu$m/s for KIF16B-powered microtubules in vitro (Hoepfner et al, 2005), which matches the speed for long-distance movement of early endosomes of ~0.30 $\mu$m/s (Gasman et al, 2003).

### KIF16B regulates Rab14-dependent fast recycling of MT1-MMP

We next tested the functional importance of KIF16B for the regulation of the surface-associated pool of MT1-MMP in macrophages. KIF16B contains an N-terminal motor domain, followed by a stalk region involved in dimerization, and a PI(3)P binding PX domain (Blatner et al, 2007; Pyrpassopoulos et al, 2017), and a short C-terminal extension (Fig 2A). Three independent siRNAs were established, leading to ~60–80% knockdowns of KIF16B, respectively (Fig 2B) (note: each time, two out of three siRNAs were used in subsequent experiments, according to commercial availability). The effect of KIF16B depletion was tested by three independent

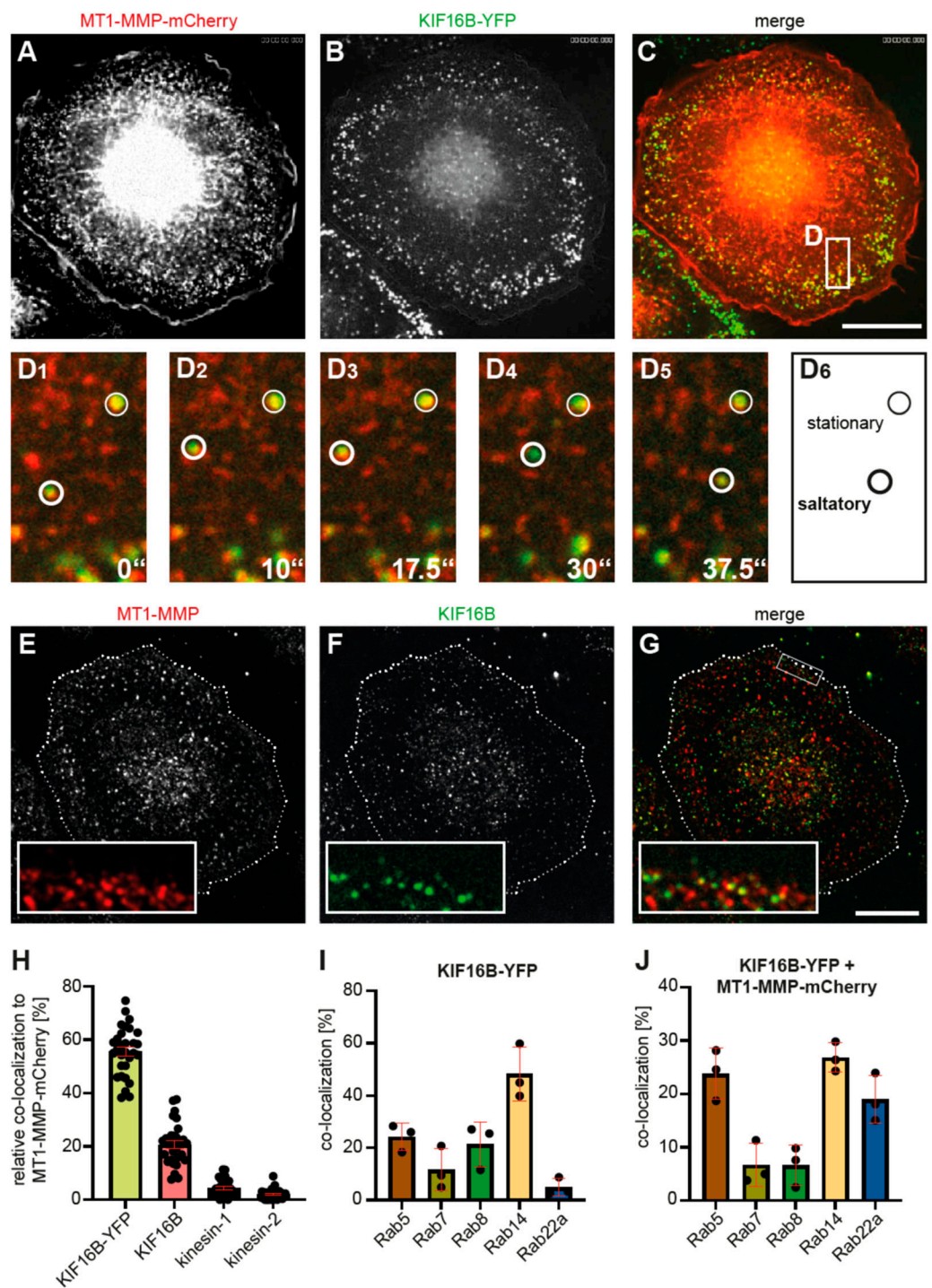

**Figure 1. MT1-MMP associates with KIF16B in a Rab14-positive pathway.**
**(A, B, C)** Still images from live cell video of primary macrophage expressing MT1-MMP-mCherry (A) and KIF16B-YFP (B), with merge (C). **(D)** White box in (C) indicates region of detail images shown in (D1–D6). Note localization of MT1-MMP-mCherry and KIF16B-YFP to both stationary (white circle) and saltatory (bold white circle) vesicles (D6). Time is indicated in sec in (D1–D5). See also Video 1. **(E, F, G)** Confocal micrographs of primary macrophage stained for endogenous forms of MT1-MMP (E) and KIF16B (F), with merge (G). **(E, F)** White box in (G) indicates region of detail images in (E, F, G). Dashed white line indicates cell circumference (note that adjacent cells, also with colocalization signals, are not marked). Scale bars: 10 μm. **(H, I, J)** Statistical evaluation of MT1-MMP-mCherry colocalization with kinesin motors: overexpressed KIF16B-YFP, or endogenous KIF16B, kinesin-1 (KHC) or kinesin-2 (KIF3A) (H), of double overlap of KIF16B with indicated RabGTPases (I), and of triple overlap of KIF16B-YFP and MT1-MMP-mCherry with indicated RabGTPases (J). N = 3 × 30 cells, from three different donors. **(H)** Note high overlap of MT1-MMP-mCherry with both endogenous and overexpressed forms of KIF16B (H). Values are given as mean ± SEM. For specific values, see Table S1.

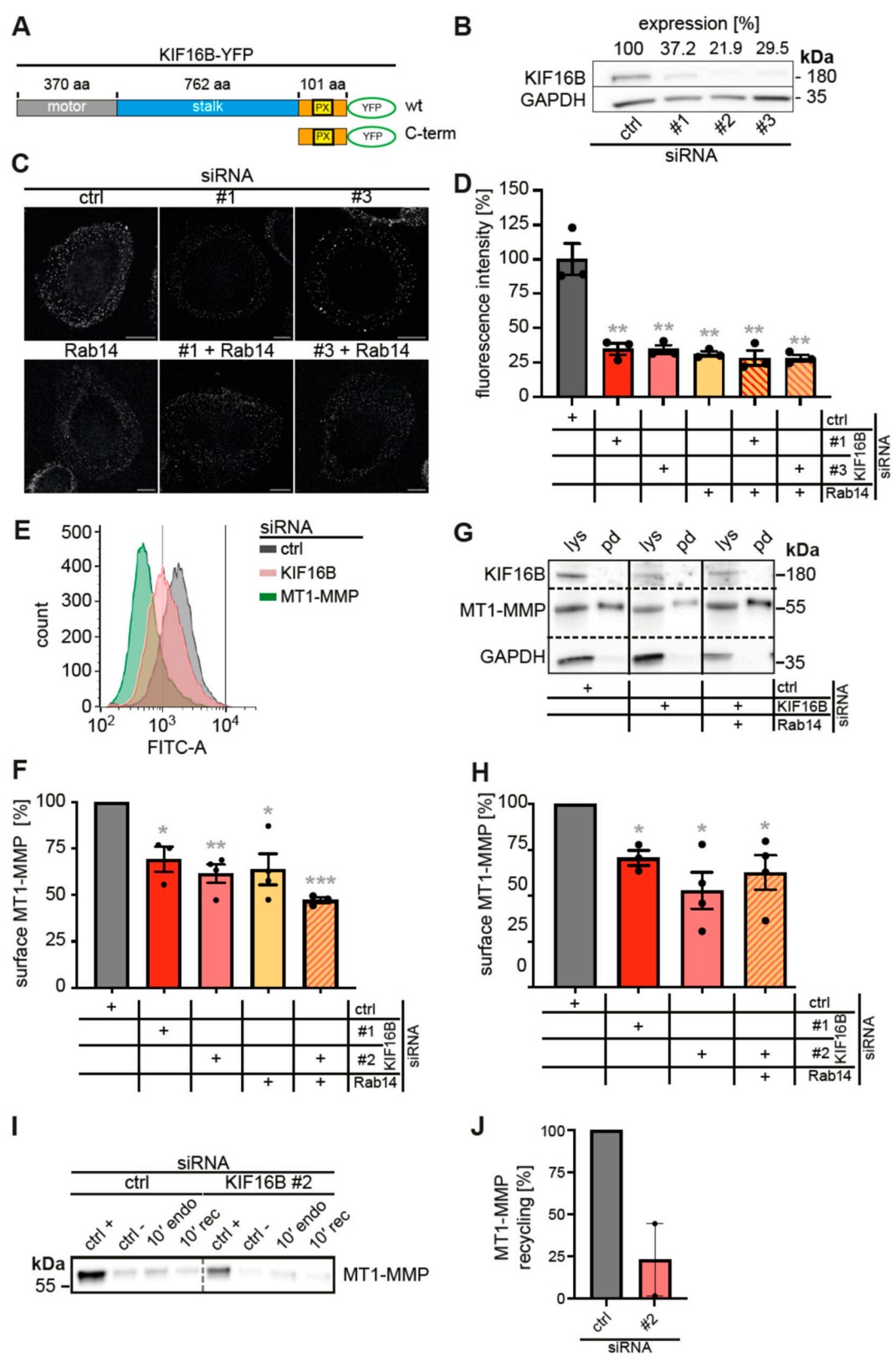

**Figure 2. KIF16B regulates MT1-MMP surface exposure.**

**(A)** Domain composition of KIF16B, with N-terminal motor domain, stalk domain, PX domain, and C-terminal extension. Respective sizes are indicated in number of aa residues. YFP-fused constructs used in this study include a WT construct and a construct consisting of the C-terminus, comprising the PX domain and the C-terminal extension (Cterm). **(B)** Efficiency of siRNA-mediated depletion of KIF16B, with mean from three independent experiments given. Western blot of lysates treated with three

methods. First, immunofluorescence staining of surface-associated MT1-MMP indicated a ~65% reduction of this MT1-MMP pool in cells treated with KIF16B siRNAs, compared to controls (Fig 2C and D). Treatment of macrophages with an established siRNA against Rab14 (Wiesner et al, 2013) led to a similar reduction (Fig 2C and D). Moreover, combined treatment with KIF16B-specific and Rab14-specific siRNAs did not show additive effects (Fig 2C and D), indicating that KIF16B and Rab14 act in the same pathway.

To test the impact of KIF16B on MT1-MMP trafficking by alternative methods, and also to quantify higher numbers of cells, we next established FACS-based measurement (Fig 2E and F) and also biotin labeling of surface-localized MT1-MMP (Fig 2G and H) in cells depleted for KIF16B. FACS-based quantification of surface-localized MT1-MMP showed that treatment with KIF16B-specific siRNAs resulted in reductions of 31% and 39%, compared to controls. Depletion of Rab14 led to a similar reduction of 36%, whereas combined knockdown of KIF16B and Rab14 resulted in a higher, but not statistically different, reduction (Fig 2F). Labeling of surface proteins with biotin and subsequent quantification of Western blots further demonstrated that cells treated with KIF16B-specific siRNA showed reductions in the surface levels of endogenous MT1-MMP of 29% and 48%, whereas combined depletion of KIF16B and Rab14 led to a reduction of 37%, again showing no additive effects (Fig 2H). To directly investigate the impact of KIF16B on MT1-MMP recycling, we next performed recycling assays using surface biotinylation. Importantly, macrophages treated with KIF16B-specific siRNA showed a ~80% reduction in MT1-MMP recycling, supporting the notion that KIF16B is the main motor driving MT1-MMP recycling in these cells (Fig 2I and J).

Collectively, these complementary experimental approaches indicate that KIF16B is a positive regulator of the surface-associated pool of MT1-MMP, and that it acts in the same pathway as Rab14. KIF16B thus emerges as a kinesin motor that drives Rab14-dependent fast recycling of endocytosed MT1-MMP back to the macrophage surface. Interestingly, depletion of KIF16B and/or Rab 14 led to stronger reductions in MT1-MMP surface levels in immunofluorescence experiments as compared with FACS and biotinylation assays. This is likely based on the fact that cells analyzed

in immunofluorescence experiments could be selected for strong depletion of proteins, whereas FACS and biotinylation assays include large numbers of cells with varying degrees of knockdown. Of note, MT1-MMP is only the fourth cargo identified for KIF16B, in addition to FGFR (Ueno et al, 2011), transferrin receptor (Perez Bay et al, 2013), and MHCI (Weimershaus et al, 2018). It should thus be worthwhile to test further transmembrane-containing cell surface proteins for their potential KIF16B-driven recycling.

### The KIF16B C-terminus regulates MT1-MMP surface levels

The C-terminus of kinesins constitutes their cargo-binding domain (Kumari & Ray, 2022), which has also been demonstrated for KIF16B (Hummel & Hoogenraad, 2021) (Fig 2A). We therefore reasoned that a construct containing the isolated C-terminus of KIF16B should be able to work as a competitive inhibitor of endogenous KIF16B, thus uncoupling MT1-MMP vesicles from KIF16B-driven recycling (Fig 3A). A respective fluorescently labeled construct (KIF16B-Cterm-YFP) was generated and expressed in primary macrophages, together with MT1-MMP-mCherry. KIF16B-Cterm-YFP was found to closely associate with MT1-MMP-mCherry–containing vesicles in both fixed (Fig 3B–E) and living cells (Fig 3F, Video 2), demonstrating that it contains the requisite regions to bind these carriers. Conversely, KIF16B-Cterm-YFP did not associate with endogenous KIF16B (Fig S2A–C), indicating that it induced dissociation of the endogenous motor from MT1-MMP vesicles.

We next tested whether uncoupling of MT1-MMP vesicles from KIF16B-driven recycling also impacted on MT1-MMP surface levels. For this, KIF16B-YFP, KIF16B-Cterm-YFP or YFP as control were expressed in primary macrophages coexpressing MT1-MMP-mCherry. Surface levels of MT1-MMP-mCherry were measured by immunofluorescence using anti-mCherry staining of unpermeabilized cells. Of note, the expression of KIF16B-YFP did not lead to significant changes in MT1-MMP surface levels. By contrast, the expression of KIF16B-Cterm-YFP led to a ~45% reduction in MT1-MMP-mCherry surface levels (Fig 3G).

The KIF16B C-terminus is able to bind Rab14 (Ueno et al, 2011). In agreement with this, we could already show that Rab14 is a critical

individual siRNA specific for KIF16B or with control siRNA, as indicated. Blots were developed with KIF16B- or GAPDH-specific antibody as the loading control, as indicated. Molecular weight in kD on the left. Expression levels of KIF16B are indicated. **(C, D)** MT1-MMP surface levels in macrophages depleted for KIF16B, Rab14 or combinations. **(C)** Fluorescence micrographs of unpermeabilized macrophages, stained for surface-associated MT1-MMP, based on Alexa568-based fluorescence. Cells were treated with KIF16B-specific siRNAs, Rab14-specific siRNA or combinations thereof, as indicated. Scale bar: 10 $\mu m$. **(D)** Fluorescence intensities of surface-associated MT1-MMP, based on Alexa Fluor-based fluorescence. Cells were treated with KIF16B-specific siRNAs, Rab14-specific siRNA or combinations thereof. Fluorescence intensities for control siRNA were set to 100%; (n = 3 × 30 cells, from three different donors). **P < 0.01, according to unpaired t test Values are given as mean ± SEM. For specific values, see Table S1. **(E, F)** Flow cytometric analysis of endogenous MT1-MMP surface levels. Macrophages were treated with MT1-MMP-, KIF16B- or Rab14–specific siRNA and stained for surface-associated MT1-MMP without fixation. Cells treated with control siRNA and stained with isotype IgG antibody were used as negative control. Histogram (E) shows results from representative experiments, respective siRNAs are indicated, out of three to four respective experiments (from at least three different donors) are shown. Values are shown as mean ± SEM. *P < 0.05, **P < 0.01, ***P < 0.005, according to one-sample t test. For specific values, see Table S1. **(G, H)** Analysis of MT1-MMP surface levels by biotinylation. Macrophage surface-associated proteins were labeled by membrane-impermeable biotin, cells were lysed, and biotinylated proteins were precipitated using streptavidin-coupled agarose beads. **(G)** Western blots of respective whole cell lysates ("lys") and pelleted beads ("pd") from cells treated with indicated siRNAs. Blots were developed using indicated antibodies, with molecular weight in kD given on the right. **(H)** Statistical evaluation of MT1-MMP surface levels, as determined by surface biotinylation. N = 3 or 4, from at least three different donors. Values represent the ratio of surface-associated MT1-MMP versus total cellular MT1-MMP, the latter normalized to GAPDH levels. Values are given as mean ± SEM. *P < 0.05, according to one-sample t test. For specific values, see Table S1. **(I, J)** Analysis of MT1-MMP recycling by biotinylation. Macrophage surface-associated proteins were labeled by membrane-impermeable biotin, treated as described, cells were lysed, and biotinylated proteins were precipitated using streptavidin-coupled agarose beads. **(I)** Western blots of respective whole cell lysates ("lys") and pelleted beads ("pd") from cells treated with indicated siRNAs. Blots were developed using indicated antibodies, with molecular weight in kD given on the right. **(J)** Statistical evaluation of MT1-MMP recycling, as determined by surface biotinylation recycling assay. N = 2, from two different donors. Values represent the ratio of recycled MT1-MMP under knockdown conditions in comparison with cells transfected with control siRNA. All samples were normalized for protein levels in the positive control measured by Ponceau staining. Values are given as mean ± SEM. For specific values, see Table S1.

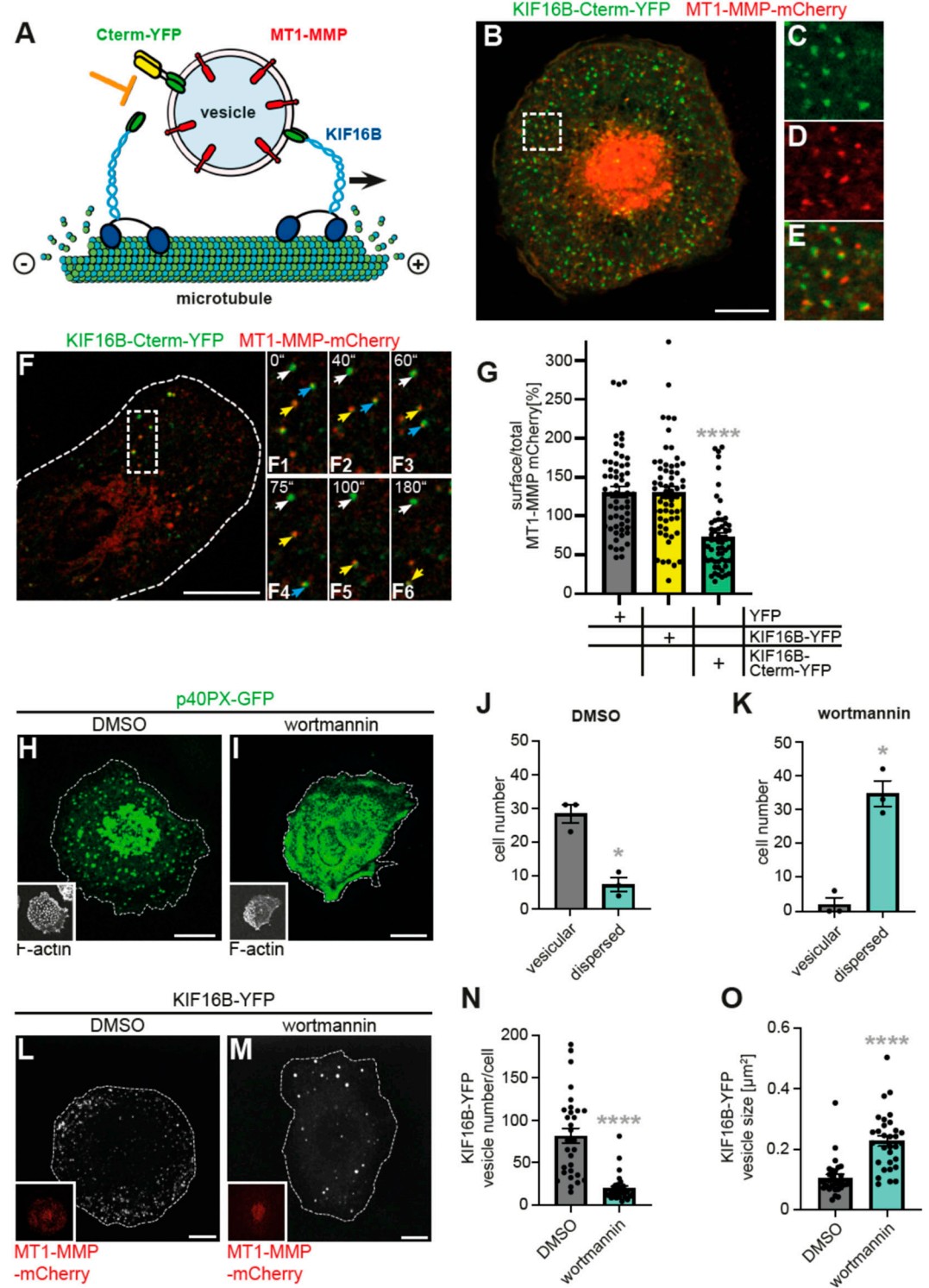

**Figure 3. The KIF16B C-terminus regulates MT1-MMP surface exposure.**
**(A)** Model of KIF16B-dependent transport of MT1-MMP vesicles along microtubules. *Right*: KIF16B, a microtubule plus end-directed motor, binds to a MT1-MMP–positive vesicle via its C-terminus, containing Rab14- and PI(3)P-binding sites. *Left*: A construct containing the isolated C-terminus fused to YFP (KIF16B-Cterm-YFP) acts as a competitive inhibitor, thus uncoupling (orange block) MT1-MMP vesicles from KIF16B-dependent transport. **(B, C, D, E)** Confocal micrograph of a macrophage expressing KIF16B-Cterm-YFP (green) and coexpressing MT1-MMP-mCherry (red), fixed sample. Scale bar: 10 μm. **(C, D, E)** Individual channels are shown in (C, D), with merge in (E). White box in (B) indicates the region shown in detail images. See also Fig S2A. **(F)** Confocal micrograph of a macrophage expressing KIF16B-Cterm-YFP (green) and coexpressing MT1-MMP-mCherry (red), still images from Video 2. Dashed white box indicates the region of still images shown in (F1–F6). Note dynamic colocalization of KIF16B-Cterm-YFP with MT1-MMP-mCherry–positive vesicles. Time since start of recording is indicated in sec. Scale bar: 5 μm. **(G)** Statistical evaluation of surface

regulator of MT1-MMP recycling by KIF16B (Fig 2C–H). In addition, the C-terminus also contains a PhoX homology (PX) domain that binds PI(3)P, enabling the motor to attach to PI(3)P-containing membranes (Blatner et al, 2007; Pyrpassopoulos et al, 2017) such as endosomes (Stenmark & Gillooly, 2001) (Fig 3A). Accordingly, we could show that KIF16B colocalizes with MT1-MMP, especially at early endosomes that are positive for Rab5a, and in vesicles in slow or fast recycling pathways that are positive for Rab14 or Rab22a, respectively (Figs 1H–K and S1B). Therefore, we next set out to determine the potential impact of PI(3)P binding on MT1-MMP trafficking by KIF16B. Macrophages expressing the PI(3)P reporter p40PX-EGFP were treated with DMSO as control or with 1 $\mu$M wortmannin to inhibit PI3 kinase. Of note, the sensor was present at vesicular structures in 75% of DMSO-treated cells (Fig 3H and J), whereas 94% of wortmannin-treated cells showed a dispersed signal (Fig 3I and K), indicating that PI(3)P was mostly absent from vesicles. Cells expressing KIF16B-YFP showed a 77% reduction in the numbers of KIF16B-YFP–positive vesicles upon wortmannin treatment (Fig 3L–N). Interestingly, the remaining KIF16B-YFP vesicles also appeared to be brighter and larger, and showed also a more peri-nuclear localization (Fig 3M), while still being motile Video 3). Indeed, size measurements showed that KIF16B-YFP vesicles had a ~2x increased size in wortmannin-treated cells, compared with controls (Fig 3O; Table S1).

As no increase in vesicle fusion was observed (Fig S2D; Video 3), this enlargement is likely based on the general swelling of early and late endocytic compartments upon wortmannin treatment, as reported for human melanoma cells (Fernandez-Borja et al, 1999). A similar trend was seen for endogenous KIF16B vesicles, with wortmannin treatment leading to a ~1.2x larger size (Fig S2E; Table S1). In line with these findings, endogenous KIF16B was found to localize at a subset of MT1-MMP-mCherry–positive, and often enlarged, vesicles (Fig S2F–H). Interestingly, FACS-based measurements of the cell surface pool of endogenous MT1-MMP in macrophages showed no alteration upon wortmannin treatment compared with controls (Fig S2I). It is currently unclear whether this is based on unaltered net recycling of the protease or on, respectively, reduced endocytosis or enhanced exocytosis.

Next, we interrogated the potential impact of Rab14 on the observed alterations in KIF16B-YFP vesicle size and localization upon wortmannin treatment. Macrophages transfected with Rab14-specific siRNA or control siRNA were treated with 1 $\mu$M wortmannin at day 3 post-transfection, and vesicle parameters were analyzed. Rab14-depleted cells showed no significant differences in vesicle size and number, compared with controls (Fig S3A and B). In contrast, wortmannin treatment of controls or Rab14-depleted cells

led to a 2.0x or 2.4x increase, respectively, in vesicle size, (Fig S3A), and a 76% or 73% reduction, respectively, in vesicle number (Fig S3B). Moreover, the peripheral localization of vesicles, typical for KIF16B overexpression (Fig 1A–C) (Hoepfner et al, 2005), was apparently unaltered in Rab14 knockdown cells and became only apparent upon wortmannin treatment (Fig S3C–H). These results indicate that, whereas PI3K inhibition led to absence of PI(3)P from vesicles, KIF16B-YFP could still associate with some vesicles that were positive for MT1-MMP (Fig 3L and M). This is likely based on the interaction with Rab14, as Rab14-GFP was still present at KIF16B-mCherry–positive vesicles upon wortmannin treatment (Fig S3I–N).

Collectively, we show that the isolated KIF16B C-terminus can act as a dominant negative construct, leading to uncoupling of MT1-MMP vesicles from the endogenous KIF16B motor and thus to a ~45% reduction of the MT1-MMP surface pool. The overexpressed KIF16B C-terminus contains binding sites for both PI(3)P and Rab14 (Ueno et al, 2011), and is thus able to bind to double-positive vesicles. The endogenous motor, outcompeted by the KIF16B C-terminal construct, should still be able to bind to vesicles that are positive for either PI(3)P or Rab14, which most likely explains the persistent vesicular localization upon KIF16B-Cterm-YFP overexpression. These results also point to a specific role of the KIF16B C-terminus in MT1-MMP recycling, which would be in line with earlier findings showing that the KIF16B PX domain cannot be replaced by the PH domain of KIF1A in vesicle binding and transport (Hummel & Hoogenraad, 2021).

## KIF16B regulates ECM degradation and 3D invasion of macrophages

To test the relevance of KIF16B-dependent fast recycling for the regulation of cell surface-associated activity of MT1-MMP, we next investigated the impact of KIF16B depletion on ECM (extracellular matrix) degradation. Macrophages were treated with KIF16B- or Rab14-specific siRNA, or a combination of both, and control siRNA, and were seeded on fluorescently labeled gelatin matrix. In this assay, matrix degradation becomes visible through localized loss of the fluorescent label (Fig 4A–D). Strikingly, treatment with KIF16B-specific siRNAs led to reductions of 40% and 46% in matrix degradation (Fig 4B and E). Consistent with previous observations (Wiesner et al, 2013), treatment of cells with Rab14-specific siRNA resulted in a reduction of 49% (Fig 4C and E), whereas combined depletion of KIF16B and Rab14 led to similar reductions of 61% and 59% (Fig 4D and E).

Importantly, the average number of podosomes per cell (Fig 4F) and their frequency distribution within a cell population (Fig 4G and

associated versus total MT1-MMP-mCherry in cells coexpressing YFP as control, full-length KIF16B-YFP or KIF1B-PX-YFP. Ratio in control cells is set to 100%. (n = 2 × 30 cells, from two different donors). Values are given as mean ± SEM. ****$P < 0.001$, according to unpaired $t$ test. For specific values, see Table S1. **(H, I, J, K)** Inhibition of PI3K by wortmannin leads to loss of PI(3)P from vesicles. **(H, I)** Confocal micrographs of macrophages expressing the PI(3)P reporter p40PX-GFP and treated with DMSO (H) or 1 $\mu$M wortmannin (I), with insets showing respective F-actin stainings. Scale bar: 10 $\mu$m. **(J, K)** Statistical evaluation of vesicular or dispersed signals of p40PX-GFP upon treatment with DMSO (J) or wortmannin (K). (n = 3 × 30 cells, from three different donors). Values are given as mean ± SEM. *$P < 0.05$, according to paired $t$ test. For specific values, see Table S1. **(L, M, N, O)** Wortmannin treatment results in fewer KIF16B-YFP-positive vesicles but unchanged MT1-MMP surface levels. **(L, M)** Confocal micrographs of cells expressing KIF16B-YFP and treated with DMSO (L) or 1 $\mu$M wortmannin (M), with insets showing respective MT1-MMP-mCherry signals (note: KIF16B-YFP/MT1-MMP-mCherry vesicles show a more peri-nuclear localization in cells treated with wortmannin. This is likely based on the absence of PI(3)P from early endosomes, resulting in a loss of KIF16B-YFP localization at this cell–peripheral vesicle subpopulation). **(N, O)** Statistical evaluation of KIF16B-YFP–positive vesicle numbers (N) or vesicle size (O) in cells treated with DMSO as control or 1 $\mu$M wortmannin (n = 3, from at least 10 cells each of three different donors). Vesicle size is indicated in $\mu m^2$. N = 3, from three different donors. Values are shown as mean ± SEM. ****$P < 0.001$, according to unpaired $t$ test. For specific values, see Table S1 (See also Video 3).

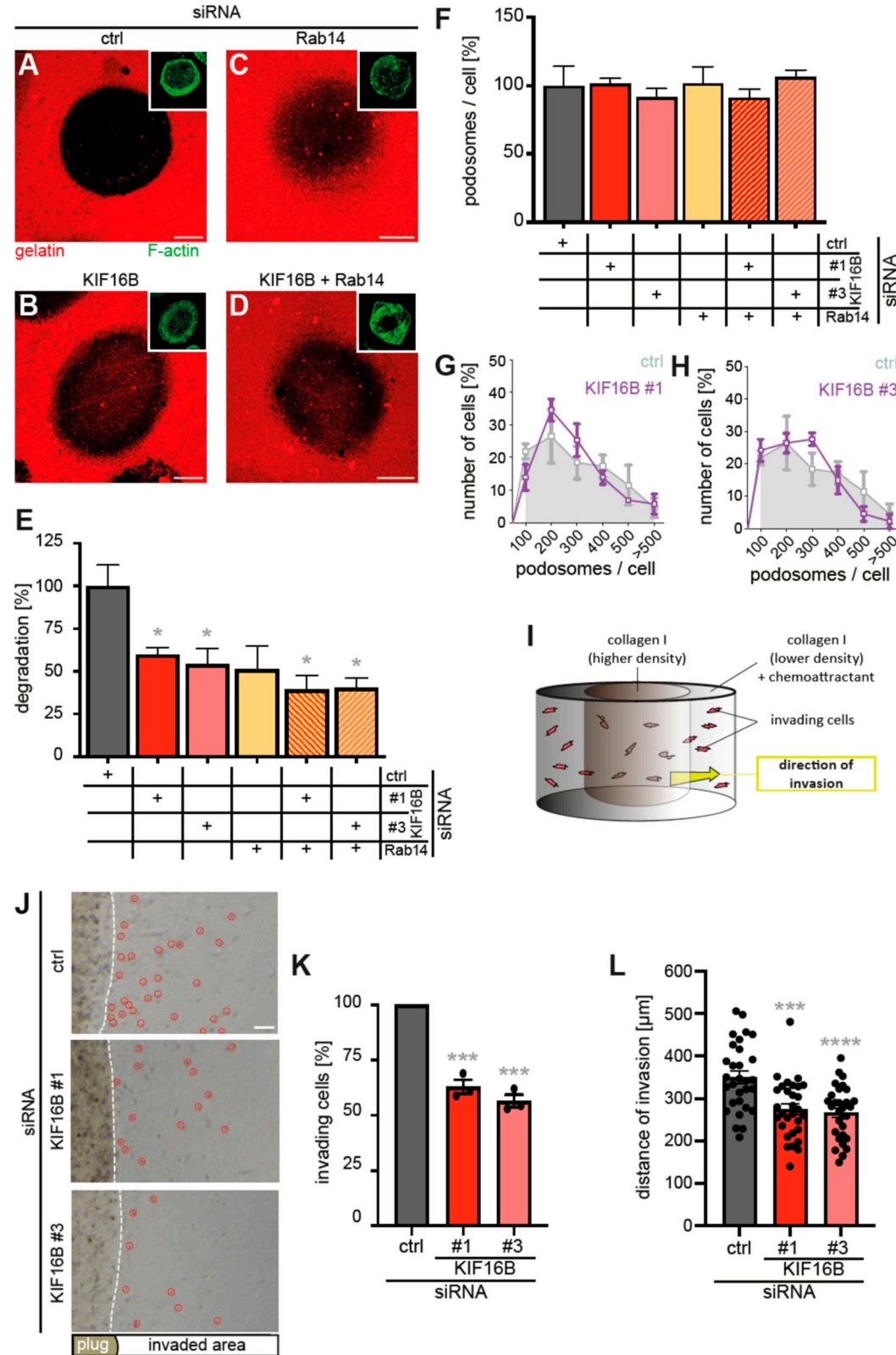

**Figure 4. KIF16B regulates matrix degradation and 3D invasion of macrophages.**
**(A, B, C, D, E)** KIF16B and Rab14 regulate ECM degradation of macrophages. **(A, B, C, D)** Confocal micrographs of macrophages treated with indicated siRNAs and seeded on Alexa568-labeled gelatin (red), and stained for F-actin (green; insets). Matrix degradation is visible as loss of the label. Scale bar: 10 $\mu$m. **(A, B, C, D, E)** Statistical evaluation of the experiments shown in A, B, C, D, with degradation in control cells set to 100%, n = 3 × 30 cells from three different donors. Values are shown as mean ± SEM,

H) were unchanged. Podosomes are the main sites of ECM degradation in macrophages (Linder et al, 2023), and MT1-MMP is considered as the "master switch" protease regulating matrix degradation at podosomes (Hey et al, 2022). The observed decrease in ECM degradation is thus not based on altered numbers of podosomes, but on their restricted degradative capability, which is likely based on lower levels of surface-associated MT1-MMP.

To assess the impact of KIF16B-dependent recycling also in a three-dimensional context, macrophage invasion was assessed in a 3D plug assay using collagen I matrix. For this, macrophages are embedded in a plug of collagen I (2.5 mg/ml), which is surrounded by a shell of less dense collagen I (2.0 mg/ml) (Wiesner et al, 2013, 2014). Invasion of cells from the plug into shell, involving surface-associated proteolysis by MT1-MMP (Wiesner et al, 2013), can be quantified using brightfield microscopy (Fig 4I). Of note, the number of invading cells was reduced by 37% and 44% in KIF16B knockdown cells, compared with cells treated with control siRNA (Figs 4J and K and S3A). Moreover, the distance of invasion, that is, the localization of individual invading cells from the rim of the plug, was reduced by 22% and 24% for KIF16B-depleted cells (Fig 4L).

Collectively, these results indicate that fast recycling driven by KIF16B impacts on the ability of macrophages to degrade matrix material in 2D, and also to invade into a 3D collagen I matrix involving proteolysis. These data appear to be in line with the description of MT1-MMP–positive podosome equivalents, so called "3D podosomes," in macrophages embedded in collagen I (Van Goethem et al, 2011; Wiesner et al, 2013). Moreover, siRNA-based depletion of MT1-MMP in this assay almost completely abrogates invasion (Wiesner et al, 2013). The observed ~40% reduction of the number of invading macrophages and their ~25% reduced distance of invasion upon depletion of KIF16B thus seems to be consistent with the ~40% reduction of MT1-MMP at the cell surface upon depletion by the motor within a larger cell population, as determined by FACS or biotinylation assays (Fig 2F and H).

Based on the data from the current study, we can now update the previously presented model (Wiesner et al, 2013) of MT1-MMP trafficking in macrophages (Fig 5A): MT1-MMP is transported to the cell surface via exocytic vesicles that are positive for Rab8a and driven by kinesin-1 and -2 (Wiesner et al, 2010, 2013). At the cell surface, MT1-MMP is present in dot-like microdomains or podosome-associated islets (El Azzouzi et al, 2016), where it can interact with and cleave surface-associated proteins or ECM components. Re-internalization of the protease proceeds via Rab5a-positive early endosomes, which can lead to degradation via Rab7-regulated late endosomes and lysosomes. However, a major part of the internalized MT1-MMP pool is recycled back to the cell surface via Rab14-dependent fast recycling or Rab22a-regulated slow recycling pathways. This recycling is mainly driven by KIF16B, which associates with MT1-MMP in Rab14- and also in Rab22a-regulated pathways. Considering that Rab14-dependent fast recycling is the predominant recycling pathway in a 3D context (Wiesner et al, 2013), and that depletion of Rab14 leads to a similar reduction of surface-associated MT1-MMP as depletion of KIF16B, KIF16B/Rab14-driven fast recycling thus emerges as a crucial trafficking route that regulates the surface-associated pool of MT1-MMP. The potential role of KIF16B in Rab22a-dependent slow recycling should merit a more detailed investigation.

### KIF16B depletion in macrophages impacts on cancer cell co-invasion

Macrophages can associate with tumors, and an increased burden of tumor-associated macrophages is coupled with a poor patient prognosis (Condeelis & Pollard, 2006). This is likely because of macrophage-based remodeling of the tumor-surrounding ECM, which enables escape of cancer cells, or on proteolytic release of factors such as CD44 from the macrophage cell surface or of VEGF from the ECM that support tumor growth and metastasis (Kessenbrock et al, 2010). Therefore, we next investigated whether the KIF16B-dependent recycling pathway in macrophages also impacts on co-invasion of cancer cells. We first screened H1299 non-small cell lung carcinoma cells, MDA-MB-211 breast cancer cells, MeWo melanoma cells, and HeLa cervix carcinoma cells for expression of MT1-MMP and for their ability to assemble into uniform spheroids. Low expression of MT1-MMP was detected in MeWo and HeLa cells, whereas MDA-MB-211 cells showed high expression, and H1299 cells intermediate levels. As H1299 cells also formed the most regular spheroids, which did not dissociate during transfer into collagen I gels, we chose a GFP-expressing H1299 for subsequent experiments (Fig 5B and C).

H1299-GFP spheroids consisting of 8,000 cells at the day of seeding were generated by cultivation in ultra-low attachment plates and embedded in 40 $\mu$l collagen I matrix (2.5 mg/ml), which also contained $8 \times 10^3$ primary human macrophages, corresponding to $2 \times 10^5$ macrophages/ml gel (Fig 5B) (note: for a test on the potential influence of macrophage numbers on tumor spheroid size and the number of invading cancer cells, see Fig S4B–D). Macrophage–tumor spheroid assays were cultured for 3 d, and growth of spheroids, and the number of tumor cells invading into the surrounding space, were evaluated by fluorescence microscopy. For this, F-actin of all embedded cells was stained by Alexa Flour 568 Phalloidin, with GFP allowing distinction between macrophages and tumor cells, and DAPI staining of nuclei enabling cell counts (Fig 5C–G).

---

*$P < 0.05$, according to unpaired $t$ test. For specific values, see Table S1. **(F)** Statistical evaluation of podosome number per cell in macrophages treated with indicated siRNAs. (n = 3 × 30, from three different donors) Values are shown as mean ± SEM. **(G, H)** Statistical evaluation of cells showing indicated number of podosomes in macrophages treated with control siRNA (gray graphs) or each of two individual KIF16B-specific siRNAs (magenta graphs). (n = 3 × 30, from three different donors). Values are shown as mean ± SEM. For specific values, see Table S1. **(I, J, K)** KIF16B regulates 3D invasion of macrophages. **(I)** Model of collagen I invasion assay. Macrophages (red objects) are embedded in a type I collagen plug (2.5 mg/ml) and invade the surrounding shell of less dense type I collagen (2 mg/ml), indicated by yellow arrow. **(J)** Brightfield micrographs of 3D invasion assays at day 4 after seeding. Dashed white lines indicates the border between collagen matrix with embedded macrophages and collagen matrix with invaded cells, visible as dark dots, and highlighted by red circles. Scale bar: 100 $\mu$m. See also Fig S4A. **(K, L)** Quantification of macrophage number (K) or distance of invasion (L), of cells treated with indicated siRNAs invading collagen I matrix, at day 4 after seeding. Values for control siRNA were set to 100%. N = 3 × 10 fields of view (K) or 3 × 10 cells (L), from three different donors. Values are shown as mean ± SEM. ***$P < 0.005$, ****$P < 0.001$, according to one-sample $t$ test. For specific values, see Table S1.

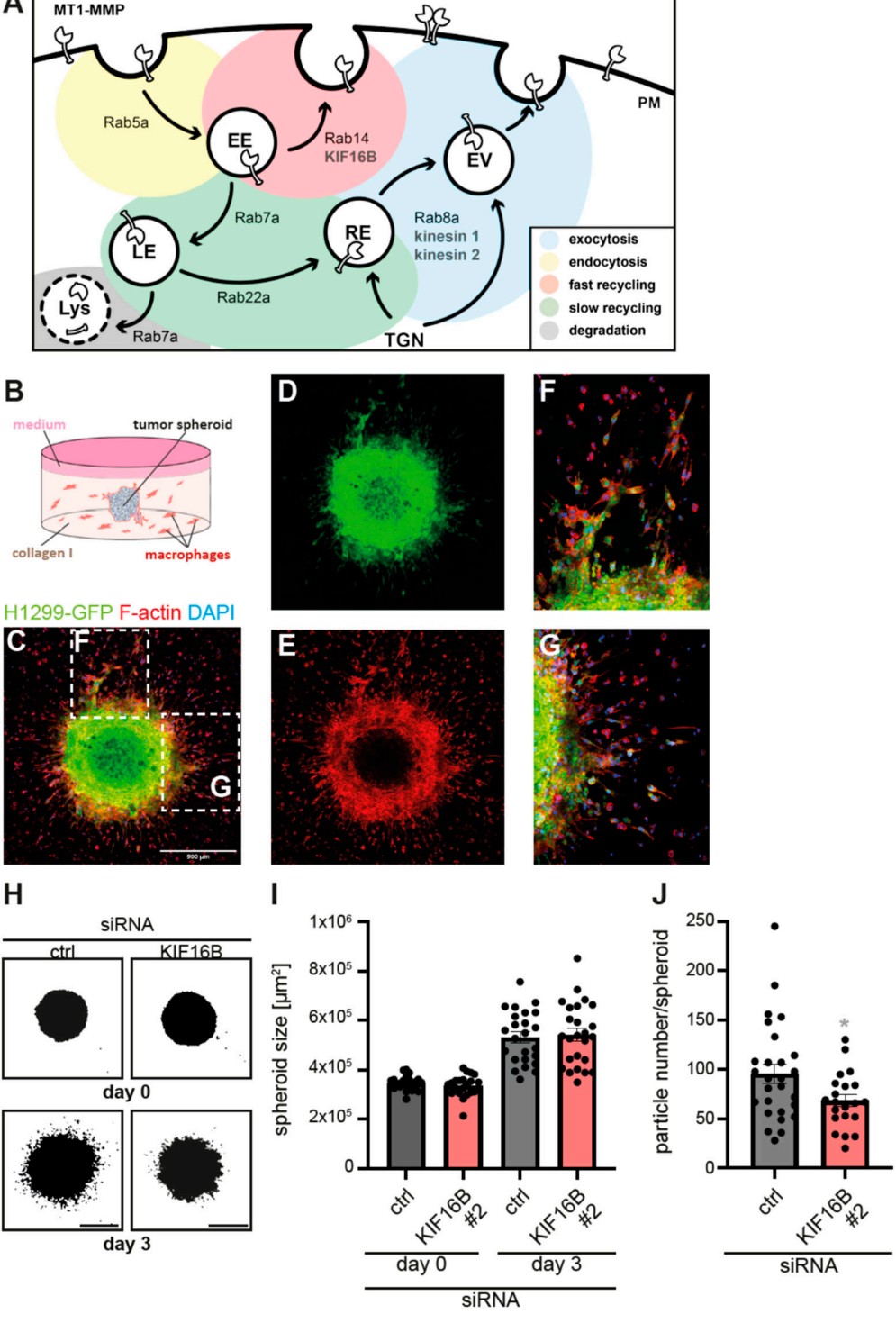

**Figure 5. KIF16B-based MT1-MMP recycling in macrophages impacts on cancer cell co-invasion.**
**(A)** Model of MT1-MMP trafficking in primary human macrophages. Intracellular trafficking pathways to and from the plasma membrane are depicted, together with known vesicle regulators of the RabGTPase family and known kinesin motors associated with respective pathways. Note the role of KIF16B in the Rab14-dependent fast recycling pathway of the proteinase back to the cell surface. For details, see text, Conclusion section. PM, plasma membrane; TGN, trans-Golgi network; EE, early endosome; EV, exocytic vesicle; RE, recycling endosome; LE, late endosome; Lys, lysosome. **(B)** Cartoon of tumor spheroid invasion assay, with central tumor spheroid (gray), collagen I matrix (light red), and cocultivated macrophages (red). **(C, D, E, F, G)** A spheroid of H1299 non-small cell lung carcinoma cells constitutively expressing GFP (H1299-GFP) embedded in collagen I matrix together with primary human macrophages. **(C, D, E)** GFP signal, (E) F-actin staining, with merge in (C). Maximum projection of a confocal z-stack. Scale bar: 500 $\mu$m. **(C, F, G)** Dashed white boxes in (C) indicate the detailed images shown in (F, G). Note both the formation of collective strands and also isolated invading cancer cells. **(H, I, J)** Cocultivation of H1299-GFP spheroids with macrophages depleted for KIF16B does not change spheroid size but number of invading cancer cells.
**(H)** Representative images of H1299-GFP spheroids cocultivated with macrophages treated with indicated siRNAs at day 0 and day 3 of the experiment. GFP signal shown in black. Maximum projection of confocal Z-stacks. Scale bar: 500 $\mu$m.
**(I, J)** Statistical evaluation of spheroid size (I) at days 0 and 3 of respective experiments or of number of invading cancer cells at day 3 (J). Cocultivated macrophages were treated with indicated siRNAs 3 d before embedding into collagen I matrix together with cancer cells. Values are shown as mean ± SEM. *$P < 0.05$, according to unpaired $t$ test. For specific values, see Table S1.

H1299-GFP tumor spheroids were cultivated in 3D collagen I in the presence of macrophages treated with control siRNA or with KIF16B-specific siRNA 3 d before setting up the assay (Fig 5B and C), with consistent knockdown efficiency during the time of the experiment monitored by a Western blot time course (Fig S4E). Tumor spheroid area increased from day 0 to day 3 of experiments, as expected, but was comparable between all setups that contained macrophages treated with the indicated siRNAs (Fig 5H and I). In contrast, the number of invading tumor cells, as determined by analysis of GFP-positive particles, was reduced by 29% in setups containing macrophages depleted for KIF16B (Fig 5H and J). We conclude from these data that the KIF16B-driven recycling pathway

in primary macrophages also impacts on the ability of H1299 cells to escape from spheroids and invade into the ECM.

These results point to the KIF16B-driven recycling pathway in macrophages as an important element of the tumor microenvironment. It is highly likely that KIF16B-driven recycling impacts not only on MT1-MMP trafficking, but also on the cell surface exposure of other cargo proteins that are involved in cell invasion and also other cellular functions such as growth factor signaling. Accordingly, depletion of KIF16B has been shown to lead to accumulation of FGFR2-positive vesicles in the perinuclear region (Ueno et al, 2011). Also, KIF16B has been shown to interact with endosomal PTPD1, a cytosolic non-receptor tyrosine kinase (Carlucci et al, 2010) that activates the EGF pathway after growth factor stimulation (Cardone et al, 2004). Analyzing the proteome of KIF16B/Rab14 vesicles will thus be a promising first step to further investigate the importance of this trafficking pathway in both immune and cancer cell invasion.

# Materials and Methods

### Cell culture

Primary human monocytes were isolated from buffy coats (kindly provided by Frank Bentzien, Transfusion Medicine, UKE, Hamburg, Germany) by centrifugation in Ficoll; 12.5 ml blood were coated on 15 ml Ficoll (PromoCell) and centrifuged for 30 min at 4°C and 460$g$. Leukocyte fractions were transferred in a new 50-ml Falcon tube and filled up to 50 ml with cold RPMI 1640 medium (Invitrogen). Cells were washed twice in RPMI 1640 and centrifuged for 10 min as described above. Enriched leukocytes were resuspended in 400 $\mu$l monocyte buffer (5 mM EDTA and 0.5% human serum albumin in DPBS, pH 7.4), mixed with 100 $\mu$l of a suspension of magnetic beads coupled to CD14 antibodies (Miltenyi Biotec) and incubated for 15 min on ice. The mixture was then loaded on a MS Separation column (Miltenyi Biotec) previously placed in a magnetic holder and equilibrated with 500 $\mu$l cold monocyte buffer. Trapped CD14$^+$ monocytes were washed on a column with 500 $\mu$l monocyte buffer and after the removal of the magnet were eluted with 1 ml monocyte buffer into 15 ml cold RPMI 1640. After centrifugation for 10 min at 4°C and 460$g$, the supernatant was removed and cells were resuspended in 40 ml RPMI 1640 and seeded on a six-well plate (Sarstedt) at a density of 10$^6$ cells per well. After adhesion of monocytes, RPMI medium was replaced by 2 ml monocyte culture medium (RPMI 1640 substituted with 15% human serum [prepared in-house] and 100 $\mu$g/$\mu$l penicillin/streptavidin). Monocytes were cultivated in an incubator at 37°C, 5% $CO_2$, and 90% humidity; every 3–4 d, the culture medium was replaced by a fresh medium. Cells were cultured in RPMI containing 20% autologous serum at 37°C, 5% $CO_2$, and 90% humidity.

### Flow cytometry

Macrophages depleted for indicated targets by specific siRNA were detached with cold DPBS containing 10 mM EDTA. Subsequently, cells were transferred into 1.5 ml tubes and centrifuged for 5 min at 300 rcf in a tabletop centrifuge (Eppendorf). The resulting pellet was washed with 1 ml cold DPBS and centrifuged again under the same conditions, followed by determination of cell numbers using a counting chamber. Cells were suspended in 100 $\mu$l staining buffer (DPBS + 1 mM EDTA + 0.5% BSA), and 5 $\mu$l (per 10$^6$ cells) of primary conjugated MT1-MMP (488 nm) (Cat.No: FAB9181G; R&D Systems) antibody were added. The cell suspension was incubated at RT for 30 min with occasional shaking. To remove unbound antibody, 1 ml DPBS was added, and cells were centrifuged again (see above). The cell pellet was resuspended in 100 $\mu$l of cold DPBS and transferred onto ice. Cell surface levels of MT1-MMP were measured using a FACS Canto II (BD Biosciences) analyzer. For this, cells were transferred into into FACS tubes and diluted with DPBS depending on the density of the cell suspension (1.0–1.5 ml).

### Generation and electroporation of plasmid DNA

Full-length KIF16B-YFP was a kind gift from Marino Zerial. p40PX-GFP was a kind gift from Michael Yaffe. MT1-MMP-pHluorin was a kind gift from Philippe Chavrier. The KIF16B-Cterm-YFP construct was cloned by PCR amplification of a 372-bp fragment of the KIF16B-YFP C-terminus using a specific KIF16B-Cterm forward primer 5′-GTTCTCGAGGACCTGATGGACCCAAT-3′ and reverse primer 5′-GTTGGATCCATGGTATGTTTCGAGAG-3′. Effectiveness of the PCR amplification was checked on a 0.7% agarose gel and the respective gel fraction was eluted and cleaned with a Gel Recovery Kit (Zymo Research). As the primer sequence contains two restriction enzyme recognition sites, the amplicon was digested with *Xho*I (New England BioLabs) (part of the forward primer) and *Bam*HI (New England BioLabs) (part of the reverse primer) and ligated into a pEYFP-N1 plasmid backbone digested with the same restriction enzymes. For the generation of Kif16b-mCherry, pmCherry-N1 vector backbone was amplified using primers that included *Xho*I and *Bsh*TI restriction sites: forward primer 5′-AAATTTCTGAGATGGGATCCGCATCGGTC-3′ and reverse primer 5′-AAATTTACCGGTGGCCCCGTCCCGTGGCTGC-3′. In parallel, full-length KIF6B was amplified with forward primer (5′-AAATTTACCGGTCGC-CACCATGGTG-3′ and reverse primer (5′-TTTAAACTCGAGATCTGAGT-CCGGTAGCGC-3′, incorporating the same restriction sites. Gel fragments were extracted and purified using a Gel Recovery Kit (QIAGEN) after the PCR amplification was verified using a 1% agarose gel. The amplicons were digested with FastDigest *Xho*I and FastDigest *Bsh*TI enzymes (Thermo Fisher Scientific), followed by ligation. Ligation products were transformed into competent *E.coli* bacteria and cultured on an agar plate supplemented with kanamycin to select for positive clones. Positive clones were tested by colony PCR and respective clones were further enriched in liquid cultures. To check for correct DNA sequence, plasmid DNA was prepared from the liquid culture and sent for sequencing.

For transfection of primary human macrophages with plasmid DNA, cells were first detached with Accutase solution, incubated for 45–60 min, and washed with DPBS. Cells were counted and suspended in R-Buffer (Invitrogen) at a concentration of 10$^6$ cells per 100 $\mu$l. For the transfection with plasmid DNA, the cells were mixed with 0.5 $\mu$g DNA per 10$^5$ cells in R-Buffer. Cells were transfected using a Neon Transfection System (Thermo Fisher Scientific) with the following settings: two pulses with 1,000 V for 40 ms.

## siRNA-mediated knockdown

For siRNA-based depletion of proteins, cells were detached using Accutase (Thermo Fisher Scientific) and incubated at 37°C and 5% $CO_2$. Detachment of cells was monitored using a light microscope, the enzymatic reaction was stopped by addition of serum-containing media (e.g., mono medium), and the cell suspension was centrifuged for 5 min at 300 rcf. The cell pellet was washed with DPBS and centrifuged again. Afterward, cells were resuspended in DPBS and counted. For each transfection, $1 \times 10^6$ cells were centrifuged and resuspended in 100 $\mu$l R-Buffer (Invitrogen). The cell suspension was added to 5 $\mu$l of 20 $\mu$M target siRNA in a 1.5-ml reaction tube and slightly mixed by pipetting the solution up and down before loading it on a Neon Transfection System (Thermo Fisher Scientific). The electroporation settings were the same as for plasmid DNA transfection. The following siRNAs were used for this study: luciferase/ctrl: 5′-AGGTAGTGTAACCGCCTTGTT-3′; MT1-MMP: 5′-AACAGGCAAAGCUGAUGCAGA-3′; KIF16B #1: 5′-GGAAAGUCAUACA-CUAUGA-3′; KIF16B #2: 5′-GAACGUCAUUUCUGCCUUA-3′; KIF16B #3: 5′-GCUUCCACAUCGAGAACAA-3′; Rab14: 5′-AAGAAGTACATATAACCACTT-3′.

## Cell-surface biotinylation

Surface biotinylation was performed using a membrane-impermeable version of biotin (Sulfo-NHS-biotin; APExBIO). Cells were first transferred onto ice and incubated for 15 min in DPBS (++) (containing 2.5 mM $CaCl_2$, × 1 mM $MgCl_2$ (pH 8.0)) (hereafter, DPBS (++)). Afterward, cells were incubated for 30 min in DPBS (++) containing 1 mg/ml of Sulfo-NHS-biotin on ice. Residual biotin was removed by washing the cells three times with ice cold DPBS (++) and quenching the remaining unbound biotin with a 100 mM glycin solution (pH 7.7). Cells were lysed immediately on ice with RIPA-buffer containing protease inhibitor cocktail (Cat.No: 04693124001; Roche) for 15 min. To separate biotinylated proteins from the whole-cell lysate, 50 $\mu$l of streptavidin agarose beads (Cat.No: 20353; Thermo Fisher Scientific) were washed with 100 $\mu$l of ddH$_2$O and pelleted. The lysate was added to beads and incubated under rotation at 4°C overnight. Samples were subsequently centrifuged at 16,873$g$ for 10 min, and the supernatant was removed. Pelleted beads were washed with 1 ml of RIPA-buffer for at least 30 min under rotation at 4°C, followed by another centrifugation step. Supernatant was removed, and 100 $\mu$l of elution buffer (10 mM biotin, 2% SDS, 6 M urea, 2 M thiourea, 100 mM NaCl, 10 mM DTT, 50 mM $Na_2HPO_4$, 1/5 SDS loading buffer (5x) in milliQ water) were added to the beads, followed by an incubation for 15 min at 95°C on a shaker. Samples were analyzed subsequently, along previously collected whole-cell lysates, by Western blot.

## Surface biotin recycling assay

Biotinylation was performed according to Cervero et al (2021), with a cleavable version of biotin (sulfo-NHS-SS-Biotin, A8005; APExBIO) for 30 min on ice. Cells were subsequently washed with ice-cold DPBS (++) and unbound biotin was quenched by washing cells three times for 10 min with ice-cold glycine containing quenching solution (100 mM glycine in DPBS (++); pH: 7.7). Biotin-labeled cells were washed again with cold DPBS (++), and the positive control was

directly lysed using RIPA-buffer. The negative control was lysed after incubating the cells three times with cold MESNA solution (100 mM in TBS buffer) to remove surface biotin and for determining the efficiency of cleavage. Next, endocytosis and recycling samples were incubated at 37°C for 10 min in the incubator and un-endocytosed surface biotin was removed by washing the cells three times with MESNA solution, with a subsequent washing step with DPBS (++), before lysing the endocytosis sample. The recycling sample was incubated again for 10 min at 37% in the incubator before removal of surface biotin by MESNA, with washing steps as described. Before lysing the cells, the sample was washed again with cold DPBS (++). Biotinylated proteins were eluted from the streptavidin agarose resin by addition of elution buffer (25% SDS 5x loading buffer, 25% DTT (1 M) and 50% RIPA buffer). All samples were analyzed by Western blot by normalizing individual signals from the blot against intensities of Ponceau S Stainings of respective lanes. The ratio of recycled protein was determined by subtracting values of reexposed protein at a time point of 20 min from values of endocytosed protein at a time point of 10 min.

## Macrophage cell lysates

To obtain macrophage cell lysates, cells were washed once with DPBS and lysed by addition of ice-cold RIPA buffer. The suspension was incubated for 10 min on ice and subsequently transferred to a 1.5 ml Eppendorf reaction tube. After centrifugation for 10 min at 16,873$g$ and 4°C, the lysate was transferred into a 1.5 ml fresh tube without disturbing the resulting pellet. After addition of SDS sample buffer, the lysates were incubated at 95°C for 10 min before analysis by Western blot. Protein lysates were generally separated in freshly prepared acrylamide gels (Acrylamide Kit, 20%; Bio-Rad TGX FastCast) or gradient gels (mPAGE 4–12% Bis–Tris, 10 × 8, 15-well; premade; Millipore) and blotted with the iBlot 2 system on pre-stacked membranes (iBlot 2 NC Mini Stacks; Invitrogen). Protein bands were recorded after staining with primary and secondary antibodies using different automated (cytiva, Amersham Image Quant 800 with auto mode and full dynamic range or LI-COR C-DiGit blot scanner) or classical developers.

## Matrix degradation assay

Measurement of degradative capabilities was performed utilizing the rhodamine–gelatin degradation assay as previously described (e.g., briefly, gelatin [swine, Carl Roth] was labeled with NHS-Rhodamine (Thermo Fisher Scientific) as established by Chen et al (1994). 12-mm glas coverslips were coated with rhodamine–gelatin and the substrate crosslinked with 0.5% glutaraldehyde (Carl Roth). The coverslips were washed with RPMI 1640 (Gibco) and culture medium to remove residual glutaraldehyde and labeled gelatine. siRNA depleted primary human macrophages were seeded on the coverslips at a density of $8 \times 10^4$ cells. After incubation of the coverslips for 6 h in the incubator, the cells were fixed with 3.7% formaldehyde before staining with Alexa Fluor 488 phalloidin. Coverslips were mounted on object slides with mowiol (Calbiochem) containing 1,4-diazabicyclo[2.2.2]octane (25 mg/ml; Sigma-Aldrich). The amount of loss in rhodamine–gelatin signal intensity was measured underneath the cell area, determined by phalloidin signal, relatively to non-degraded area using ImageJ

software. Control siRNA treated cells were used as a control and set as 100% of degradation. To compare the intensities, laser intensities were unchanged within samples.

### 3D plug-invasion assay

To measure 3D invasion of primary human macrophages, a two-phase plug invasion assay consisting of collagen-I (collagen type I, rat tail, Corning) with different densities was performed, according to Wiesner et al (2014). First, the inner collagen core was generated by polymerization of collagen-I (2.5 mg/ml) containing macrophages treated with specific siRNA in an empty 96-well plate in the incubator. Afterward, the polymerized plug was transferred into an empty 24-well plate and arranged in the center of the well. The second phase was generated by slow addition of another collagen-I mix (2 mg/ml) laced with 10 ng/ml macrophage colony-stimulating factor (M-CSF, Relia Tech) to stimulate migration. After polymerization, cell culture media were added, and the setup was incubated at 37°C and 5% $CO_2$. Analysis of invasive capability of the macrophages was performed after 4 d by taking images of the border of the inner collagen-I plug and counting of cells that migrated into the lighter collagen phase.

### Tumor spheroid co-invasion assay

For generation of tumor spheroids, H1299-GFP cells were detached with Trypsin-EDTA (0.25%, phenol red; Gibco), washed with DPBS and counted. Uniform spheroids were generated from 8,000 cells diluted in 25 $\mu$l cancer cell media (DMEM + 1% pen–strep + 10% FBS + 0.1 mM nonessential amino acids + 2 mM l-glutamine) and incubated in BIOFLOAT ultra-low attachment plates (Sarstedt) for 3 d. Solid spheroids were washed by addition of 300 $\mu$l DPBS and transferred with a 1 ml pipette tip into a 15-well $\mu$-slide 3D (ibidi). Residual liquid was removed, and spheroids were embedded in 40 $\mu$l collagen I (2.5 mg/ml) containing 8 × $10^3$ siRNA-depleted macrophages. Chambers were incubated for 30 min at 37°C and 5% $CO_2$ in the incubator to allow for uniform polymerization of collagen I. Subsequently, 25 $\mu$l cancer cell media were added onto the collagen I, and cells were imaged at indicated time points with a Leica DMi8 microscope (Leica DMi8 with a TCS SP8 AOBS confocal point scanner equipped with a 10x HC PL APO CS, NA 0.40 objective and Leica LAS X SP8 software). Quantification was performed using ImageJ2 (Version: 2.3.0/1.53 s). For the measurement of tumor spheroid area and particle numbers, the fluorescence images were z-projected and adjusted by first applying the Huang auto-threshold method implemented in ImageJ (Image > Adjust > Threshold > Huang > Auto). Afterwards, the resulting noise was reduced by Despeckle (Process > Noise > Despeckle) before selecting the spheroid main area with the wand tool to measure it. The selected area was depleted after the measurement and the remaining signal quantified by particle analysis (Analyze > Analyze particles …).

### 96-well tumor-spheroid co-invasion assay

To test the influence of different amounts of macrophages on the invasion of single tumor cells from a spheroid, a 96-well collagen I assay was established. 8 × $10^3$ H1299-GFP cells in 25 $\mu$l cancer cell medium were seeded in a 96-well ultra-low attachment plate (Sarstedt #83.3925.400; BIOFLOAT) and incubated at 37°C and 5% $CO_2$ for 3 d. Subsequently, spheroids were checked for uniform formation. Different cell numbers of primary human macrophages (0–1 × $10^5$ cells, see text) were suspended in rat tail collagen I mix (final concentration in the well: 2.5 mg/ml, Corning) and added directly into the wells containing 25 $\mu$l medium and the spheroid. For mixing collagen I and the medium, the plate was incubated for 10 min on ice on a rocker and subsequently transferred to the incubator for 30 min, to allow polymerization of the collagen. To each well, 100 $\mu$l of cancer cell culture medium was added. Spheroids were imaged 3 d later using an Nikon Eclipse TiE with 10x CFI Plan Fluor DL Phase (NA: 0.3; WD [mm]: 15.2; pixel size: 1.1 $\mu$m). Analysis was performed with a specially written Fiji macro to detect single cells, after z-projection, thresholding and particle analysis, in the vicinity of spheroids.

### Immunofluorescence microscopy

Macrophages were seeded at a density of $10^5$ cells per glass coverslip (12 mm diameter) and fixed for 10 min in 3.7% formaldehyde, washed three times in PBS, and permeabilized for 10 min in PBS containing 0.1% TritonX-100. After three washes with PBS, cells were incubated for 30 min in blocking solution (2% BSA in PBS), washed briefly in PBS, and incubated for 60 min in the primary antibody solution. Cells were washed three times in PBS and then incubated for 30 min in a secondary antibody solution. After three washes in PBS, coverslips were mounted on glass slides with Mowiol 4–88 (Carl Roth) containing p-phenylenediamine (Sigma-Aldrich). Images of fixed samples were acquired with a confocal laser-scanning microscope (Leica DMi8 with a TCS SP8 AOBS confocal point scanner equipped with an oil-immersion 63× HC PL APO Oil CS2 NA 1.40 objective and Leica LAS X SP8 software or a Visitron spinning disk microscope using a SoRa (CSU W-1 SoRa; Yokogawa) disk for super resolution (microscope Nikon Eclipse TiE, 60x Apo TIRF [corr.] Oil objective, NA: 1.49, WD [mm]: 0.13 [CS: 0.10–0.22 at 23°C or 37°C], SORA disk: 0.065, camera: Photometrics Prime 95B [back-illuminated sCMOS, 11 μm pixel-size, 1,200 × 1,200 pixels]). The following antibodies were used for IF staining: mouse anti-MT1-MMP (MAB3328; Merck), mouse anti-KIF16B (ab67790; Abcam), rabbit anti-Rab5 (Cell Signaling), mouse anti-Rab14 (Abcam), rabbit anti-Rab22a (ProteinTech), mouse anti-Rab8a (Sigma-Aldrich), rabbit anti-Rab7 (Cell Signaling), anti-KIF3A (Clone 28; BD Biosciences), anti-KHC (clone SUK-4; Covance).

### Co-localization analysis

Co-localization was measured with an object-based method described by Moser et al (2017). Before the analysis of fluorescence images, vesicle properties like size and geometrical descriptors were analyzed to establish a standardized size range. The macro was adjusted by the determined descriptors, and different auto thresholds were tested. The number of overlapping objects was set in relation to the total number of

detected objects (e.g., number of KIF16B vesicles overlapping with MT1-MMP–positive vesicles versus number of MT1-MMP–positive vesicles) to determine the amount of co-localizing objects and translated into percentages. In addition to the object based method, the co-localization of each Rab protein (as in Fig 1J) was calculated in respect to the amount of co-localization between MT1-MMP-mCherry and KIF16B-YFP by using the Image J co-localization plug in.

### Quantification of vesicle number and size

Vesicle number and size were analyzed using Fiji. For this, a region of interest around the cell was drawn using the freehand selection tool. After background subtraction, an auto threshold was applied, and the remaining signals were analyzed with the Analyze Particles function. Settings for control cells and conditions were kept constant to allow direct comparison.

### Wortmannin assays

To determine the effects of wortmannin on vesicle populations, macrophages were first transfected with plasmid DNA (p40PX-GFP, KIF16B-YFP, MT1-MMP mCherry), as described before, and seeded in eight-well coverslips ($\mu$-Slide eight well; ibidi). After overnight expression of the constructs, the cells were washed twice with DPBS and incubated for 1 h in 1 ml RPMI supplemented with 1 μM wortmannin or DMSO as control. Cells were fixed after removal of the media with 3.7% formaldehyde for 15 min at RT and covered with DPBS until imaging with a Visitron spinning disk microscope using a SoRa (CSU W-1 SoRa; Yokogawa) disk for super resolution (microscope Nikon Eclipse TiE, 60x Apo TIRF [corr.] Oil objective, NA: 1.49, WD [mm]: 0.13 [CS: 0.10–0.22 at 23°C or 37°C], SORA disk: 0.065, camera: Photometrics Prime 95B [back-illuminated sCMOS, 11 μm pixel-size, 1,200 × 1,200 pixels]) or a Leica DMi8 (TCS SP8 AOBS confocal point scanner equipped with a 63x HC PL APO Oil CS2 objective [NA: 1.4, WD (mm): 0.14] and 3x HyD, 2x PMT, 1x Trans-PMT detectors with Leica LAS X SP8 software). Imaged cells were analyzed for vesicle numbers and size as described above. To determine the surface levels of MT1-MMP after wortmannin treatment, macrophages were seeded at a density of $10^6$ cells in six-well chambers and washed with DPBS after attachment. Cells were incubated for 1 h in RPMI containing 1 $\mu$M wortmannin or DMSO as a control, and detached with DPBS + EDTA (10 mM) also containing 1 $\mu$M wortmannin. Staining and FACS analysis were performed as described.

### Measurement of podosome numbers

Podosome numbers were quantified after fixation and F-actin staining using a Fiji-based macro (Cervero et al, 2013).

### Statistical analysis

The graphs and statistical analyses were created/arranged using GraphPad Prism 9 and Microsoft Excel. Datasets were analyzed by paired/unpaired $t$ test, one-sample $t$ test or ordinary one-way ANOVA. Significance values were labeled in the following way: $*P < 0.05$, $**P < 0.01$, $***P < 0.001$, $****P < 0.0001$.

## Supplementary Information

## Acknowledgements

We thank Frank Bentzien (UKE Transfusion Medicine) for buffy coats, Philippe Chavrier for MT1-MMP-pHluorin, Marino Zerial for KIF16B-YFP, Michael Yaffe for p40PX-GFP, Andrea Mordhorst for expert technical assistance, the UKE microscopy facility (umif) for help with microscopy and image analysis, and Martin Aepfelbacher for continuous support. This work is part of the doctoral thesis of S Hey and was supported by Deutsche Forschungsgemeinschaft (CRC877/B13). We acknowledge financial support from the Open Access Publication Fund of UKE—Universitätsklinikum Hamburg-Eppendorf and DFG—German Research Foundation.

### Author Contributions

S Hey: formal analysis, validation, investigation, visualization, and methodology.
C Wiesner: formal analysis, validation, investigation, visualization, and methodology.
B Barcelona: resources.
S Linder: conceptualization, resources, formal analysis, supervision, funding acquisition, validation, visualization, methodology, project administration, and writing—original draft, review, and editing.

### Conflict of Interest Statement

The authors declare that they have no conflict of interest.

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
