## [Reviewer comments · Life Science Alliance]

Life Science Alliance

KIF16B drives MT1-MMP recycling in macrophages and promotes co-invasion of cancer cells

Sven Hey, Christiane Wiesner, Bryan Barcelona, and Stefan Linder

DOI: <https://doi.org/10.26508/lsa.202302158>

Corresponding author(s): *Stefan Linder, University Medical Center Hamburg-Eppendorf*

Review Timeline:

Submission Date:	2023-05-15
Editorial Decision:	2023-05-17
Revision Received:	2023-08-21
Editorial Decision:	2023-08-28
Revision Received:	2023-08-29
Accepted:	2023-08-30

Scientific Editor: *Eric Sawey, PhD*

Transaction Report:

Please note that the manuscript was previously reviewed at another journal and the reports were taken into account in the decision-making process at *Life Science Alliance*.

Reviews

Reviewer #1 Review

Comments to the Authors (Required):

The authors show that KIF16B, a Kinesin-3 family motor, plays an important role in MT1-MMP transport to the cell surface in the human primary macrophage, and shows its relevance in cancer cell invasion using in vitro spheroid model. Thus, for the first time, the manuscript reveals machinery that could support the fast recycling of the protease in cancer-associated macrophages. However, this observation in light of the past few reports offers limited novelty. It was already known that Rab14 is important for MT1-MMP recycling (Wiesner et al., 2013). It was also shown in a separate study that KIF16B via its C-terminal region interacts with Rab14 in a nucleotide-dependent manner to recycle FGFR (Ueno et al., 2011). Thus, the current study is an extension based on the earlier observations in studying the role of the motor in the transport of a cancer-associated cargo molecule (MT1-MMP).

Major points associated with the results:

(i) Fig. 2 related: The specific role of the motor in Rab14-driven MT1-MMP transport could be further addressed by including the other recycling Rabs, such as Rab22a or Rab4 in the study.

(ii) Fig. 3 related: It is better to study the role of the C-terminal region of the motor in MT1-MMP transport by rescue approach. It is not very clear why the overexpression of the C-terminal region of KIF16B leads to two distinct sets of punctae (Supple Fig. 2A)? Does it suggest that either PI3P or Rab14 individually can recruit KIF16B on the endosomal vesicles? Does motor binding to Rab14 is more important than PI3P binding? Further dissection of the C-terminal region (such as just the PX domain) may help in understanding the mechanism.

Does interaction with Rab14 and PI3P both required for the motor to be associated with the vesicles, the data does not suggest the same.

The PI3K inhibition experiment should be carried out using Immunofluorescence (for endogenous rather than overexpression of the motor). Can the authors provide co-localization of MT1-MMP with KIF16B in the presence of wortmannin? There is no discussion why in the presence of Wortmannin, larger vesicles were observed (Fig. 3M). Can the authors provide quantification for the size of the endogenous KIF16B vesicles and the KIF16B-YFP vesicles? Is it because of enhanced fusion activity driven by the motor? Surprisingly, the motility of the enlarged vesicles is not compromised. Is there any quantification of the speed or other motor-related parameters? Are the enlarged vesicles contain PI3P or Rab14, If not how KIF16B is being recruited?

It appears from the figure(Fig. 3 L,M) that the peri-nuclear MT1-MMP-cherry population is enhanced in wortmannin-treated cells. However, the authors show that the surface pool of the protease remains unchanged. It seems counter-intuitive. In light of the role of PI3P in the endocytic pathway, it is surprising to note that there is no effect of PI3K inhibition on the recycling of the protease.

(iii) Video 2. The MT1-MMP-cherry and KIF16B c-ter YFP vesicles are moving toward the nucleus. The c-terminal KIF16B does not contain a motor domain and the authors have suggested that the c-terminal region displace the Full-length motor. Discuss what ensures the motility of those vesicles?

Reference

1. Wiesner C, El Azzouzi K, Linder S. A specific subset of RabGTPases controls cell surface exposure of MT1-MMP, extracellular matrix degradation and three-dimensional invasion of macrophages. *J Cell Sci.* 2013 Jul 1;126(Pt 13):2820-33. doi: 10.1242/jcs.122358. Epub 2013 Apr 19. PMID: 23606746.
2. Ueno H, Huang X, Tanaka Y, Hirokawa N. KIF16B/Rab14 molecular motor complex is critical for early embryonic development by transporting FGF receptor. *Dev Cell.* 2011 Jan 18;20(1):60-71. doi: 10.1016/j.devcel.2010.11.008. PMID: 21238925.

Reviewer #2 Review

Comments to the Authors (Required):

KIF16B drives MT1-MMP recycling in macrophages and promotes co-invasion of cancer cells

Hey et al.

This manuscript reports the involvement of KIF16B and Rab14 in the exocytosis of recycled MT1-MMP and their potential role in enhancing cancer cell invasion. The participation of KIF16B and Rab14 in MT1-MMP exocytosis seems to be a solid finding, but they are not striking enough to justify publication as a "Report". The manuscript lacks conclusive data showing that KIF16B is solely involved in the exocytosis of recycled MT1-MMP. Also, the model of the co-invasion of macrophages with cancer cells needs more characterizations and justifications of the system.

Specific point

1. The authors showed that the knockdown of KIF16B significantly reduced but did not completely block the cell surface expression of endogenous and exogenous mCherry-tagged MT1-MMP. However, there is still a possibility that KIF16B is involved in the exocytosis of newly synthesized MT1-MMP partially, and the authors have not ruled out this possibility. It has been shown that clathrin-dependent endocytosis of MT1-MMP is cytoplasmic tail-dependent. If KIF16B is solely involved in the exocytosis of recycled MT1-MMP, the cytoplasmic domain-mutant of MT1-MMP that replaces LLY with AAA was shown to be clathrin-dependent endocytosis-defective, does not associate with KIF16B. Another mutant can lack the C-terminal three amino acids, which has been suggested to be defective in recycling. The effect of these mutations on KIF16B association with MT1-MMP-containing vesicles needs to be examined.
2. Another experiment to support the above finding is to carry out a recycling assay. There are several assay methods available, including one utilizing cell surface biotinylation. If KIF16B knockdown inhibited the recycling of MT1-MMP, this would strongly support the finding. Without these supportive experiments, the conclusion authors made has not been proven.
3. I have a problem with the analysis of the collagen invasion assay (Fig 4J). The authors evaluated the number of invading cells. This is all or none, and since KIF16B knockdown "reduced" the cell surface level of MT1-MMP by 30-50%, it is expected to affect the effectiveness of invasion but not inhibit invasion. Thus, it would make more sense to measure the distance of invasion. Previously, the impact of lack of endocytosis on the invasion ability of cells has not been clearly demonstrated. Thus, it would be very informative to examine this.
4. The authors attempted to demonstrate the potential role of macrophages (TAM) in cancer cell invasion in tumor tissue with the coculture assay of tumour spheroids and macrophages (Fig5 and 6). This is a similar assay system with a coculture of fibroblasts and cancer cells, demonstrating the collective migration of cancer cells. It has been shown that fibroblasts lead the cancer cell invasion by locating the tip of invading cancer cells. Thus, this experiment replaces fibroblasts with macrophages to see if macrophages can enhance collagen invasion of cancer cells. I believe. This is the first demonstration of the co-invasion of cancer cells with macrophages in a 3D collagen matrix, and it needs more detailed analyses. I wonder if this experimental system reflects the potential role of macrophages in tumor tissue. The authors need justification.
5. It is not described if macrophages enhance cancer invasion from the spheroids. This is crucial data to demonstrate. Second, the cell number ratio of cancer cells (8000/spheroid) vs macrophages (8000/gel) is 1:1. Providing that cancer cell invasion is enhanced by the presence of macrophages in the gel, is it dependent on the number of macrophages present in the gel?
6. The experiments would also benefit from including the following conditions. 1) macrophages silenced for mmp14 gene; and 2)

macrophages silenced for rab14 gene.

7. The images at the time 0 would help data interpretation. It would show the density of macrophages in the gel

8. What would happen if macrophages were incorporated within a cancer cell spheroid and subjected to invasion assay?

May 17, 2023

Re: Life Science Alliance manuscript #LSA-2023-02158-T

Prof Stefan Linder
University Medical Center Eppendorf
Institute for Medical Microbiology
Martinistr. 52
Hamburg 20246
Germany

Dear Dr. Linder,

Thank you for submitting your manuscript entitled "KIF16B drives MT1-MMP recycling in macrophages and promotes co-invasion of cancer cells" to Life Science Alliance. We'd invite further consideration of this manuscript at LSA pending the following revisions:

- Address Reviewer 1's Major points #2 & 3.
- Address Reviewer 2's Specific points #2, 3, 5 & 7.

Thank you for this interesting contribution to Life Science Alliance. We are looking forward to receiving your revised manuscript.

Sincerely,

- A letter addressing the reviewers' comments point by point.
- An editable version of the final text (.DOC or .DOCX) is needed for copyediting (no PDFs).
- High-resolution figure, supplementary figure and video files uploaded as individual files: See our detailed guidelines for preparing your production-ready images, <https://www.life-science-alliance.org/authors>
- Summary blurb (enter in submission system): A short text summarizing in a single sentence the study (max. 200 characters including spaces). This text is used in conjunction with the titles of papers, hence should be informative and complementary to the title and running title. It should describe the context and significance of the findings for a general readership; it should be written in the present tense and refer to the work in the third person. Author names should not be mentioned.
- By submitting a revision, you attest that you are aware of our payment policies found here: <https://www.life-science-alliance.org/copyright-license-fee>

B. MANUSCRIPT ORGANIZATION AND FORMATTING:

Reviewer #1

Major points associated with the results:

2A). Fig. 3 related: It is better to study the role of the C-terminal region of the motor in MT1-MMP transport by rescue approach.

We agree with the reviewer that a rescue approach would add complementary data. We did try this initially, by siRNA-depletion of KIF16B, followed by overexpression of an siRNA-insensitive deletion construct lacking the C-terminus, but this approach resulted in very few viable cells. Further enrichment of cells by FACS was also not possible, as cells did not reattach to coverslips or culture flasks after this procedure. We hope that the current data showing that i) a construct containing the isolated KIF16B C-terminus localizes to MT1-MMP-mCherry vesicles (Fig. 3B-F) and ii) leads to dislocation of the endogenous motor from vesicles (Suppl. Fig.2A), resulting in iii) strong reduction of MT1-MMP-mCherry surface levels (Fig. 3G), are sufficient proof for the role of the KIF16B C-terminus in MT1-MMP transport.

2B) It is not very clear why the overexpression of the C-terminal region of KIF16B leads to two distinct sets of punctae (Supple Fig. 2A)? Does it suggest that either PI3P or Rab14 individually can recruit KIF16B on the endosomal vesicles? Does motor binding to Rab14 is more important than PI3P binding? Further dissection of the C-terminal region (such as just the PX domain) may help in understanding the mechanism. Does interaction with Rab14 and PI3P both required for the motor to be associated with the vesicles, the data does not suggest the same.

To address this point, we have now performed additional experiments in Rab14 knockdown or control cells, treated with wortmannin or DMSO as control. KIF16B is a plus-end directed motor, and its overexpressed form KIF16B-YFP is mostly present in the cell periphery, indicating motor activity (Hoepfner et al., 2005). Interestingly, this localization is not altered upon Rab14 knockdown, but only upon depletion of PI(3)P, following wortmannin treatment. We conclude that binding to Rab14 is not important for peripheral localization of KIF16B, although it likely directs the motor to the Rab14-positive recycling compartment. PI(3)P binding, on the other hand, appears to be essential for peripheral localization. Respective micrographs are now shown as the new (Suppl. Fig. 3C-H,) and the results are mentioned in the text (p.10).

The overexpressed KIF16B-C-terminus contains binding sites for both PI(3)P and Rab14 (Ueno et al., 2011) and is thus able to bind to double-positive vesicles. The endogenous motor, outcompeted by the KIF16B-C-term construct from these double-positive vesicles, would still be able to bind to vesicles that are positive for either PI(3)P or Rab14, which most

likely explains the persistent vesicular localization upon KIF16B-Cterm overexpression. This point is now also mentioned in the text (p. 10). (See also points 2D,E)

2C) The PI3K inhibition experiment should be carried out using Immunofluorescence (for endogenous rather than overexpression of the motor). Can the authors provide colocalization of MT1-MMP with KIF16B in the presence of wortmannin?

We have now performed immunostainings of endogenous KIF16B in cells expressing MT1-MMP-mCherry and treated with wortmannin. We find colocalization of endogenous KIF16B at MT1-MMP vesicles, some of which are clearly enlarged, which seems to be consistent with the general swelling of the endocytic compartment upon wortmannin treatment (see also point 2D). This is now also mentioned in the text (p.9) and representative micrographs are shown as the new Suppl. Fig. 3F-H.

2D) There is no discussion why in the presence of Wortmannin, larger vesicles were observed (Fig. 3M). Can the authors provide quantification for the size of the endogenous KIF16B vesicles and the KIF16B-YFP vesicles? Is it because of enhanced fusion activity driven by the motor? Surprisingly, the motility of the enlarged vesicles is not compromised. Is there any quantification of the speed or other motor-related parameters? Are the enlarged vesicles contain PI3P or Rab14, If not how KIF16B is being recruited?

Performing live cell imaging of KIF16B-YFP-overexpressing cells, we do not observe enhanced fusion of these vesicles (gallery in new Suppl. Fig. 2D). A likely explanation for the increased size of KIF16B/KIF16B-YFP vesicles is the general swelling of early and late endocytic compartments upon wortmannin treatment, as reported for human melanoma cells (Fernandez-Borja et al., Curr Biol., 1999). This point is now also discussed in the text (p. 9), and the respective paper has been added to the reference list.

We are now providing size quantification for KIF16B-YFP vesicles, and also for endogenous KIF16B-positive vesicles, either treated with wortmannin or DMSO as control. Also in these experiments, we find that KIF16B vesicles are clearly enlarged (~1,2x) in wortmannin-treated cells, if not to the same degree (~2x) as in KIF16B-YFP overexpressing cells. Respective bar diagrams are now shown as the new Fig. 3O and Suppl. Fig. 3A,B. Please note that we had to prepare new samples for the quantification of vesicle size in the new Fig. 2O. For consistency, we thus also reevaluated vesicle number (Fig. 2N) based on these samples.

We do not detect an alteration in the speed of the remaining KIF16B-YFP vesicles. But, given that the motor is still attached, and likely also active, at this vesicle subpopulation, this would not be expected.

The enlarged vesicles do not contain PI(3)P, as this is mostly dispersed upon wortmannin treatment (see Fig. 3H-K). To address the question of potential colocalization with Rab14, we have now generated a KIF16B-mCherry construct and coexpressed it with Rab14-GFP and treated cells with wortmannin. Respective confocal micrographs show that both constructs colocalize at vesicles in control cells, as expected, but also at the remaining and enlarged vesicles in wortmannin treated cells. We conclude that the presence of KIF16B at vesicles upon wortmannin treatment is likely based on its interaction with Rab 14. This is now also mentioned in the text (p.10), and respective micrographs are shown as the new Suppl. Fig. 3I-N. (See also point 2B)

2E) It appears from the figure (Fig. 3 L,M) that the peri-nuclear MT1-MMP-cherry population is enhanced in wortmannin-treated cells. However, the authors show that the surface pool of the protease remains unchanged. It seems counter-intuitive. In light of the role of PI3P in the endocytic pathway, it is surprising to note that there is no effect of PI3K inhibition on the recycling of the protease.

Thank you for spotting the more peri-nuclear localization of MT1-MMP-mCherry/KIF16B-YFP vesicles. We have now re-examined all images and can confirm this altered localization as a consequence of wortmannin treatment (See also Suppl. Fig. 3F). Absence of PI(3)P from early endosomes also seems to be consistent with a loss of MT1-MMP localization at this vesicle subpopulation, which localizes mostly to the cell periphery. We have also added an explanatory sentence to the legend of Fig. 3M (see also point 2B).

It is indeed surprising that we find no effect of wortmannin treatment on MT1-MMP surface levels in the FACS analysis (Note: This quantification was performed for endogenous MT1-MMP, with endogenous levels of KIF16B. Unfortunately, this was not explicitly mentioned in the original manuscript. The respective panel has now been moved to Suppl. Fig. 2I, alongside additional results gained from cells with endogenous protein levels). We have therefore tried to perform cell surface biotinylation experiments (see Fig. 2G,H) also in the presence of wortmannin. However, this assay involves repeated washing and detachment steps. Upon wortmannin treatment, cells invariably detached/failed to re-attach in this assay. We are now explicitly stating that the cell surface level of endogenous MT1-MMP is not significantly altered under wortmannin treatment and that it is currently unclear whether this is based on unaltered net recycling of the protease or on, respectively, reduced endocytosis or enhanced exocytosis. (p.10).

3) Video 2. The MT1-MMMP-cherry and KIF16B c-ter YFP vesicles are moving toward the nucleus. The c-terminal KIF16B does not contain a motor domain and the authors have suggested that the c-terminal region displace the Full-length motor. Discuss what ensures the motility of those vesicles?

We have previously shown that kinesin-1 and kinesin-2 drive anterograde, exocytic transport of MT1-MMP, while dynein powers retrograde transport (Wiesner et al., Blood, 2010). It is highly likely that these motors are still present at MT1-MMP vesicles after KIF16B displacement and drive their respective movement. We have now added a short paragraph to the legend of Suppl. Video 2 to explain this.

Reviewer #2

Specific point

2. Another experiment to support the above finding is to carry out a recycling assay. There are several assay methods available, including one utilizing cell surface biotinylation. If KIF16B knockdown inhibited the recycling of MT1-MMP, this would strongly support the finding. Without these supportive experiments, the conclusion authors made has not been proven.

Thank you for this good suggestion. We have now performed recycling assays using surface biotinylation (n=3). We find that KIF16B knockdown cells show a strong reduction by ~80% of MT1-MMP recycling, compared to controls, supporting the notion that KIF16B is the main motor driving MT1-MMP recycling. Respective data are now shown as the new Fig. 2I,J and

are discussed in the text (p.8). The recycling assay has been added to the Materials and Methods section.

3. I have a problem with the analysis of the collagen invasion assay (Fig 4J). The authors evaluated the number of invading cells. This is all or none, and since KIF16B knockdown "reduced" the cell surface level of MT1-MMP by 30-50%, it is expected to affect the effectiveness of invasion but not inhibit invasion. Thus, it would make more sense to measure the distance of invasion. Previously, the impact of lack of endocytosis on the invasion ability of cells has not been clearly demonstrated. Thus, it would be very informative to examine this.

Thank you for this good suggestion. We have now also measured the distance of invasion, i.e. localization of individual cells from the rim of the plug at day 4 of the experiment. We find that the distance of invasion is also reduced by 22% and 24% for either KIF16-specific siRNA, compared to controls. This information has now been added to the text (pp. 10,11) and a respective diagram is now shown as the new Fig. 4L.

5. It is not described if macrophages enhance cancer invasion from the spheroids. This is crucial data to demonstrate. Second, the cell number ratio of cancer cells (8000/spheroid) vs macrophages (8000/gel) is 1:1. Providing that cancer cell invasion is enhanced by the presence of macrophages in the gel, is it dependent on the number of macrophages present in the gel?

The number of macrophages (8000/gel, corresponding to 2×10^5 /ml) was originally chosen as we detected no alteration of spheroid size. We have now performed a more systematic analysis of the potential influence of macrophage numbers on spheroid size and the number of invading cancer cells, adding 4×10^4 , 8×10^4 , 2×10^5 or 5×10^5 macrophages/ml. We do not detect a change statistically significant changes from controls either in spheroid size. However, 5×10^5 macrophages/ml led to a ~20% reduction in spheroid diameter, which is accompanied by a statistically significant ~2.3x increase in the number of invading cancer cells. We conclude from these data that macrophages can indeed increase the number of invading cancer cells, but only at numbers around 5×10^5 /ml or higher. Respective micrographs and statistical analyses have been added as the new Suppl. Fig. 4B-D, and the results are mentioned in the text and also in the Materials and Methods section. Please note that we are now also including more data points for the analysis of spheroid size (Fig. 5I) and number of invading cancer cells (Fig. 5J).

7. The images at the time 0 would help data interpretation. It would show the density of macrophages in the gel

We have now added B/W images of tumor spheroids also for day 0 of the experiment (new panels in Fig. 5H). For assessing the density of macrophages in the gel, please see the new Suppl. Fig. 4B which shows brightfield images of spheroids at day 0, cocultivated with different numbers of macrophages/ml.

August 28, 2023

RE: Life Science Alliance Manuscript #LSA-2023-02158-TR

Prof. Stefan Linder
University Medical Center Hamburg-Eppendorf
Institute for Medical Microbiology
Martinistr. 52
Hamburg 20246
Germany

Dear Dr. Linder,

Thank you for submitting your revised manuscript entitled "KIF16B drives MT1-MMP recycling in macrophages and promotes co-invasion of cancer cells". We would be happy to publish your paper in Life Science Alliance pending final revisions necessary to meet our formatting guidelines.

- please use the [10 author names, et al.] format in your references (i.e., limit the author names to the first 10)
- we encourage you to revise the figure legends for figure S2 such that the figure panels are introduced in an alphabetical order

A. FINAL FILES:

B. MANUSCRIPT ORGANIZATION AND FORMATTING:

****It is Life Science Alliance policy that if requested, original data images must be made available to the editors. Failure to provide**

original images upon request will result in unavoidable delays in publication. Please ensure that you have access to all original data images prior to final submission.**

The license to publish form must be signed before your manuscript can be sent to production. A link to the electronic license to publish form will be sent to the corresponding author only. Please take a moment to check your funder requirements.

Sincerely,

August 30, 2023

RE: Life Science Alliance Manuscript #LSA-2023-02158-TRR

Prof. Stefan Linder
University Medical Center Hamburg-Eppendorf
Institute for Medical Microbiology
Martinistr. 52
Hamburg 20251
Germany

Dear Dr. Linder,

Thank you for submitting your Research Article entitled "KIF16B drives MT1-MMP recycling in macrophages and promotes co-invasion of cancer cells". It is a pleasure to let you know that your manuscript is now accepted for publication in Life Science Alliance. Congratulations on this interesting work.

DISTRIBUTION OF MATERIALS:

Again, congratulations on a very nice paper. I hope you found the review process to be constructive and are pleased with how the manuscript was handled editorially. We look forward to future exciting submissions from your lab.

Sincerely,
